# *Magical*: Medical Lay Language Generation via Semantic Invariance and Layperson-tailored Adaptation

**Weibin Liao**♣♠†, **Tianlong Wang**♣†, **Yinghao Zhu**♢,
**Yasha Wang**♣♠∗, **Junyi Gao**♡§∗, **Liantao Ma**♣♠∗

♣National Engineering Research Center For Software Engineering, Peking University
♠School of Computer Science, Peking University
♢School of Computing and Data Science, The University of Hong Kong
♡Centre for Medical Informatics, University of Edinburgh
§Health Data Research UK

✉ {liaoweibin, tianlong.wang}@stu.pku.edu.cn, malt@pku.edu.cn
⚫ https://github.com/tianlwang/Magical.git

## Abstract

Medical Lay Language Generation (MLLG) plays a vital role in improving the accessibility of complex scientific content for broader audiences. Recent literature to MLLG commonly employ parameter-efficient fine-tuning methods such as Low-Rank Adaptation (LoRA) to fine-tuning large language models (LLMs) using paired expert-lay language datasets. However, LoRA struggles with the challenges posed by multi-source heterogeneous MLLG datasets. Specifically, through a series of exploratory experiments, we reveal that standard LoRA fail to meet the requirement for semantic fidelity and diverse lay-style generation in MLLG task. To address these limitations, we propose *Magical*, an asymmetric LoRA architecture tailored for MLLG under heterogeneous data scenarios. *Magical* employs a shared matrix $A$ for abstractive summarization, along with multiple isolated matrices $B$ for diverse lay-style generation. To preserve semantic fidelity during the lay language generation process, *Magical* introduces a `Semantic Invariance Constraint` to mitigate semantic subspace shifts on matrix $A$. Furthermore, to better adapt to diverse lay-style generation, *Magical* incorporates the `Recommendation-guided Switch`, an externally interface to prompt the LLM to switch between different matrices $B$. Experimental results on three real-world lay language generation datasets demonstrate that *Magical* consistently outperforms prompt-based methods, vanilla LoRA, and its recent variants, while also reducing trainable parameters by 31.66%.

## 1 Introduction

Medical Lay Language Generation (MLLG) [1] shared task surrounds the **abstractive summarization** and **lay-style generation** of biomedical articles, aiming to generate readable, accessible summaries for non-expert audiences [2]. This task addresses a critical barrier: the highly specialized language used in clinical and biomedical literature often limits public understanding and reduces patient adherence to medical advice in clinical settings. By translating complex medical content into lay-friendly language, MLLG promotes equitable access to life-related knowledge [3–5], supporting the broader

---

† Equal Contribution. ∗ Corresponding Authors.

39th Conference on Neural Information Processing Systems (NeurIPS 2025).

societal right to understand and engage with health information [6–8]. To achieve this goal, recent literature [9, 10, 2] has explored the use of parameter-efficient fine-tuning (PEFT) [11] methods such as LoRA [12] on large language models (LLMs) [13–15] by leveraging paired expert-lay language data, to enable LLMs to equip domain-specific knowledge and transformation patterns relevant to MLLG tasks. Given the scarcity of paired expert-lay language data, recent efforts [16, 17] have shifted toward utilizing **multi-source MLLG datasets** to enrich training signals and promote generalization.

**Key Insights of LoRA on MLLG.** LoRA [12] modifies the original model by injecting a low-rank decomposition into the weight updates, expressed as $y = W_0 x + BAx$, where $W_0$ denotes the pre-trained weight matrix, and $A$ and $B$ are low-rank matrices capturing the adaptation. This formulation reveals a structural resemblance to auto-encoder-based representation editing [18–21], where $W_0 x$ represents the original representation, and $BAx$ can be interpreted as an additive, low-dimensional edit in the representation space [22]. Applying LoRA to MLLG implicitly assumes that both **abstractive summarization** and **lay-style generation** can be modeled within a low-rank subspace. However, this assumption raises two critical questions: **Does the low-rank adaptation reliably preserve semantic fidelity?** and **Can they adapt robustly to diverse lay-style generation proposed by multi-source MLLG datasets?** Given that semantic fidelity is a potential optimization objective and heterogeneity is an inherent characteristic of multi-source datasets, it remains uncertain whether LoRA's rank-constrained updates can adequately model both **abstractive summarization** and **diverse lay-style generation**. This motivates a deeper investigation into the representational capacity of LoRA in scenarios requiring both semantic stability and diverse stylistic transformation.

In this study, we conduct a series of exploratory experiments using LoRA to fine-tune LLMs on multi-source MLLG datasets. Our in-depth analysis of heterogeneity in MLLG datasets and LoRA's mechanics yields several insightful observations and leads to the formulation of key hypotheses. Firstly, *despite serving the same MLLG tasks, datasets from different sources exhibit tremendous heterogeneity.* This heterogeneity further contributes to LoRA's suboptimal performance. Results indicate that rather than utilizing multiple datasets to fine-tune a single LoRA, it is more effective to employ multiple smaller LoRAs, each fine-tuned exclusively on a specific dataset. This suggests that *the detriments of data heterogeneity outweigh the potential benefits of increased training diversity.* Furthermore, we investigate semantic fidelity. Results demonstrate that *LoRA's low-rank projection causes detrimental semantic subspace shift*, which challenges the optimization objective of semantic fidelity in MLLG tasks. Based on these observations, we contend that **standard LoRA fail to meet the requirement for semantic fidelity and diverse lay-style generation in MLLG task**.

To address these challenges, we propose *Magical* (Medical Lay Language Generation via Semantic Invariance and Layperson-tailored Adaptation), an asymmetric LoRA architecture with a shared matrix $A$ for **abstractive summarization** and multiple isolated matrices $B$ for **diverse lay-style generation**. Specifically, for matrix $A$, *Magical* first employs `Semantic-Relevant Layer Identification` to identify the Transformer layers within the LLM that are semantically relevant, then applies `Semantic Contrastive Learning` to encourage matrix $A$ to project input representations into a semantic latent subspace, thereby ensuring semantic fidelity within the low-rank projection. For matrix $B$, *Magical* utilizes multiple isolated matrices $B$ to project representations into diverse lay-style subspaces, enabling adaptation to different sources of MLLG datasets individually. Furthermore, inspired by the divide-and-conquer principle [23], *Magical* incorporates the `Recommendation-guided Switch`, an external interface that prompts LLM to dynamically switch among different $B$ matrices.

Our contributions are summarized as follows:

1. **Insightly**, we provide valuable insights into existing approaches that fine-tune LLMs using LoRA for the MLLG task. Based on these insights, we conducted a series of exploratory experiments and confirmed that LoRA falls short in achieving semantic fidelity and diverse lay-style generation. To address this challenge, we propose *Magical*, a method that incorporates `Semantic Invariance Constraint` and `Layperson-tailored Adaptation`.

2. **Technically**, we design `Semantic-Relevant Layer Identification` and employ `Semantic Contrastive Learning` on matrix $A$ to enforce semantic fidelity during the low-rank projection process. Furthermore, we introduce multiple isolated matrices $B$ along with a `Recommendation-guided Switch` to enable diverse lay-style generation.

3. **Experimentally**, we conduct extensive experiments to validate the robustness of *Magical* in maintaining semantic fidelity and its effectiveness in supporting diverse lay-style generation.

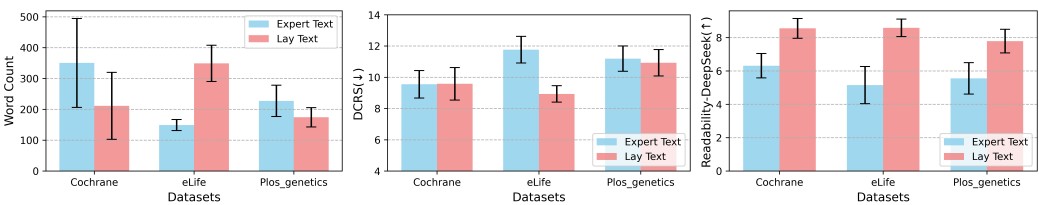

Figure 1: Distribution of Word Count, DCRS [1] and readability evaluation of DeepSeek-V3 on three heterogeneous MLLG Datasets.

## 2 Preliminary and Motivation

### 2.1 Low-Rank Adaptation (LoRA)

LoRA [12] introduces a parameter-efficient fine-tuning strategy that achieves performance comparable to full fine-tuning across various benchmarks. Instead of updating the full set of pretrained model weights $W_0$, LoRA keeps these weights frozen and injects trainable low-rank decomposition matrices into each layer of the model. Specifically, for each layer, LoRA introduces two consecutive low-rank matrices $A$ and $B$ to model the residual weight updates, thereby enabling task-specific adaptation. The process can be mathematically formulated as follows:

$$y' = y + \Delta y = W_0 x + BAx \tag{1}$$

where $y \in \mathbb{R}^d$ denotes the output and $x \in \mathbb{R}^k$ represents the input. The matrices $B \in \mathbb{R}^{d \times r}$ and $A \in \mathbb{R}^{r \times k}$ are low-rank projections, with $r \ll \min(d, k)$.

### 2.2 Heterogeneous MLLG Datasets

In this empirical study, we conduct a detailed investigation of three real-world publicly available datasets used in our work: Cochrane [24], eLife [25], and Plos_genetics [2]. Specifically, we analyze both the expert and lay texts in these datasets using three metrics: Word Count, Dale-Chall Readability Score (DCRS) [1] and readability evaluation of DeepSeek-V3 (For details of these metrics, please refer to Appendx. B). As shown in Fig. 1, we observe that for Cochrane and Plos_genetics, the word count decreases after lay transformations, whereas for eLife, the word count increases. In terms of DCRS, only eLife exhibits improved readability after lay transformations, whereas the readability of Cochrane and Plos_genetics decreases following the same process. Furthermore, based on readability evaluation conducted using DeepSeek-V3 [26], we observe that although all three datasets show an overall improvement in readability after lay transformations, the magnitude of improvement varies across datasets.

Our views the following: each MLLG dataset may be associated with an inaccessible *inter-annotator agreement* [9], resulting in distinct dataset characteristics, including simplification strategies, depth of simplification, domain-specific conventions, and stylistic preferences. In some scenarios, MLLG involves removing non-essential content while preserving the core message (word count ↓). In others, it requires supplementing the original text with additional background knowledge to explain domain-specific technical terms (word count ↑). Based on this, we derive our first key observation:

***Observation I:*** *Different medical lay language generation datasets yield diverse lay-style generation proposed by heterogeneity, stemming from differences in inter-annotator agreement.*

### 2.3 LoRA Meets Heterogeneous MLLG Datasets

**Single-LoRA *vs.* Multi-LoRA.** In this preliminary experiment, we primarily investigate whether LoRA can adapt to multi-source heterogeneous data and enable diverse lay-style generation. To achieve this goal, we focus on LoRA and conduct a series of experiments, as shown in Table. 1, to gain deeper insight into its underlying mechanisms. We adopt two independent experimental setups. In the first setup, we use a single LoRA with a rank of 24 to jointly fine-tune on three datasets: Cochrane [24], eLife [25], and Plos_genetics [2]. In the second setup, we use three separate LoRAs,

Table 1: Performance of LoRA and mLoRA (multiple smaller LoRAs) on Cochrane, eLife and Plos_genetics datasets. $n$ is the number of LoRAs, $r$ denotes the rank of each LoRA. All evaluation metrics are defined such that higher values indicate better performance ($\uparrow$).

| Methods | $r \times n$ | #Params | Cochrane | | | | eLife | | | | Plos_genetics | | | |
|---|---|---|---|---|---|---|---|---|---|---|---|---|---|---|
| | | | R-1 | R-2 | R-L | BLEU | R-1 | R-2 | R-L | BLEU | R-1 | R-2 | R-L | BLEU |
| *LLaMA3.2-3B-Instruct* | | | | | | | | | | | | | | |
| LoRA | 24×1 | 36M | 39.43 | 16.07 | 37.21 | 10.20 | 42.32 | 11.55 | 40.26 | 5.62 | 41.04 | 12.42 | 38.35 | 6.33 |
| mLoRA | 8×3 | 36M | **40.01** | **16.33** | **37.71** | **10.37** | **46.69** | **13.48** | **44.57** | **7.36** | **44.92** | **13.62** | **41.42** | **8.80** |
| *LLaMA3.1-8B-Instruct* | | | | | | | | | | | | | | |
| LoRA | 24×1 | 62M | **41.32** | **17.45** | **38.72** | 11.84 | 48.25 | 14.59 | 46.27 | 8.49 | 41.98 | 12.85 | 39.17 | 6.79 |
| mLoRA | 8×3 | 62M | 40.19 | 16.56 | 37.53 | **12.06** | **49.40** | **14.88** | **47.29** | **8.53** | **47.55** | **16.34** | **43.98** | **11.54** |
| *Qwen2.5-7B-Instruct* | | | | | | | | | | | | | | |
| LoRA | 24×1 | 60M | 41.32 | 17.45 | 38.72 | 11.84 | **48.25** | **14.59** | **46.27** | **8.49** | 41.98 | 12.85 | 39.17 | 6.79 |
| mLoRA | 8×3 | 60M | **44.23** | **19.37** | **41.60** | **14.50** | 47.38 | 13.90 | 45.42 | 7.93 | **46.41** | **16.29** | **42.95** | **9.94** |

each with a rank of 8, to fine-tune the LLM individually on each dataset. The total number of trainable parameters remains the same across both setups.

The experimental results show that, in the vast majority of cases, deploying multiple smaller LoRAs yields better performance than using a single larger one. We attribute this to the fact that while a single LoRA can leverage a larger amount of training data, its ability to adapt is constrained by the heterogeneity of the MLLG data. As a result, it struggles to finish diverse lay-style generation. This analysis yields another critical observation:

***Observation II:*** *The interference caused by the heterogeneity of data when using fully shared LoRA parameters outweighs the potential benefits brought by the additional information gained from data augmentation.*

**Semantic shift in LoRA.** A potential optimization objective in the MLLG task is ensuring semantic fidelity [2], as misinterpretation or distortion of medical information can lead to patients developing inaccurate health perceptions or making inappropriate decisions [3, 6, 27]. We conduct an in-depth analysis of semantic conveyance in low-rank projection of LoRA. Specifically, we randomly select a Multi-layer Perceptron (MLP) and project the representations of the original expert texts and the LoRA-finetuned lay texts onto the first two singular directions of a semantically relevant subspace. We then plot their respective Kernel Density Estimation (KDE) distributions [18], as shown in Fig. 2. It can be observed that the original expert texts and generated lay texts exhibit significant distributional divergence in the semantic subspace. This indicates that LoRA lacks a mechanism to preserve semantic fidelity in low-rank projection, leading to notable semantic shift during LoRA fine-tuning. The case study provided in Appendix. E.10 also demonstrates the semantic degradation caused by the naive application of LoRA. This motivates our third key observation:

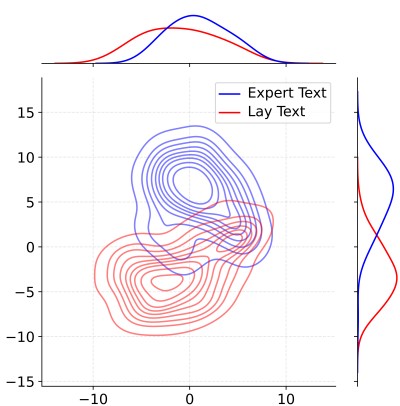

Figure 2: Projections of activations from expert text and lay text on the top-2 singular directions of the semantic subspace, which form the x- and y-axes of the KDE plot.

***Observation III:*** *The naive application of LoRA in the MLLG task demonstrates insufficient control over semantic fidelity, resulting in harmful semantic shifts that degrade model performance.*

**Summary.** Building on the above insightful observations, our goal is to more effectively leverage mixed heterogeneous data by adapting to diverse lay-style generation while preserving semantic fidelity throughout the lay transformations process. Inspired by works such as HydraLoRA [28], we propose *Magical*. *Magical* employs `Semantic Invariance Constraint` on $A$ to enforce semantic fidelity during the low-rank projection process (addressing ***Observation III***), and intro-

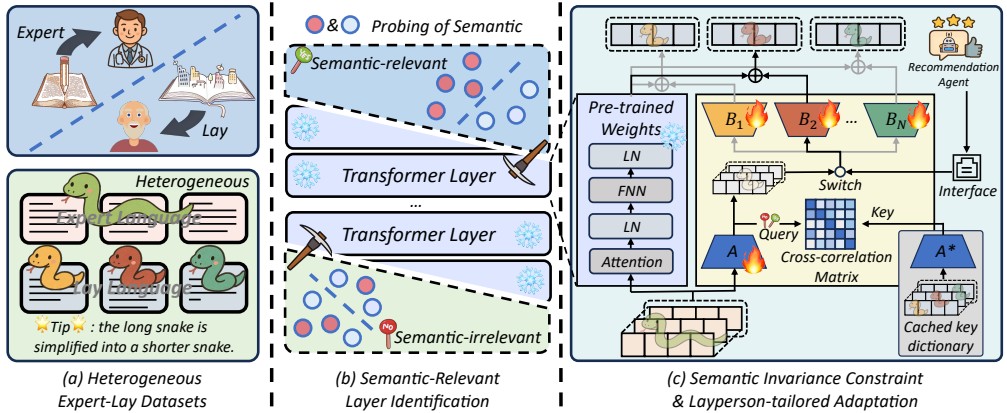

Figure 3: Overview of the *Magical*. (a) Illustrates the target audience of expert-lay language and the heterogeneity of multi-source datasets. (b) Depicts *Magical* employs probing techniques to identify semantic-relevant layers for subsequent `Semantic Contrastive Learning`. (c) Shows *Magical* applies `Semantic Contrastive Learning` on matrix $A$ to enforce semantic invariance, and utilizes an external *Recommendation Agent* to switch between different matrices $B$ for `Layperson-tailored Adaptation`.

duces `Layperson-tailored Adaptation` on $B$ to enable diverse lay-style generation (addressing ***Observations I*** and ***II***).

## 3 Methodology

### 3.1 Framework of *Magical*

We present *Magical*, a novel framework inspired by HydraLoRA [28] that employs asymmetric structure for MLLG. The framework operates on a collection of datasets $\mathcal{D} = \{\mathcal{D}_1, \mathcal{D}_2, \ldots, \mathcal{D}_N\}$, where each $\mathcal{D}_i$ corresponds to training data for a specific lay-style and $N$ is the number of datasets.

*Magical* utilizes a shared matrix $A \in \mathbb{R}^{r \times k}$ for abstractive summarization and multiple isolated matrices $B = \{B_1, B_2, \ldots, B_N\}$, where each $B_i \in \mathbb{R}^{d \times r}$ is dedicated to a specific lay-style generation. The parameter $r$ represents the low-rank dimension, $k$ is the input dimension, and $d$ is the output dimension. The process can be mathematically formulated as follows:

$$y = W_0 x + \sum_{i=1}^{N} \alpha_i \cdot B_i A x \tag{2}$$

where $\alpha_i$ is a branch control variable that controls the degree of branch participation.

As illustrated in Fig. 3, *Magical* further employs `Semantic Invariance Constraint` on matrix A and `Layperson-tailored Adaption` on matrix B to enable more robust medical lay language generation.

### 3.2 Semantic Invariance Constraint on $A$

**Semantic-Relevant Layer Identification.** Prior work [29] has shown that different layers of LLMs capture distinct functional representations. Accordingly, *Magical* employs probing techniques [30–32] to identify layers that are more semantically relevant and applies customized LoRA fine-tuning specifically to those layers. To achieve this, *Magical* utilizes expert–lay language pairs with aligned semantics and determines the semantic relevance of each layer based on its probing accuracy on a semantic classification task.

Specifically, for the paired expert–lay language dataset $\mathcal{D} = \{(x_o^{(1)}, x_s^{(1)}), \cdots, (x_o^{(N)}, x_s^{(N)})\}$, where $x_o$ and $x_s$ denote the expert language (original) and the lay language (simplified) respectively, and

$N$ represents the number of samples in the dataset. *Magical* constructs a semantic consistency classification task by aggregating expert and lay language, and creates a probing dataset $\mathcal{D}^* = \{[x_o^{(i)}; x_s^{(j)}], \cdots\}$. A pair is considered a positive sample if $i = j$, and a negative sample otherwise. Let $x^* = [x_o^{(i)}; x_s^{(j)}]$, *Magical* defines the probe $p_l(x^*) = \text{Sigmoid}(\langle \theta_{0 \to l}, x^* \rangle)$ for each layer $l$ of the LLM to detect the semantic-relevance of the activations, where $\theta_{0 \to l}$ denotes the parameters of the first $l$ layers of the LLM.

Next, *Magical* randomly splits the dataset into a 4:1 ratio of training to testing sets and fits a binary linear classifier $p(\cdot)$ on the training set. The layers achieving the top-$K$ validation accuracy are identified as semantic-relevant layers, and subsequent `Semantic Contrastive Learning` is applied specifically to these layers.

**Semantic Contrastive Learning.** Based on the earlier ***Observation III***, *Magical* encourages LoRA to project input representations through matrix $A$ into a semantic-relevant low-rank subspace, in order to preserve semantic fidelity during the low-rank projection process. To achieve this, contrastive learning [33] is applied to representations in this space. Specifically, *Magical* seeks to establish a clear boundary between samples with different semantics in the semantic space, and contrastive learning is a well-established approach for this purpose [18, 34]. For the expert–lay language dataset $\mathcal{D} = \{(x_o^{(1)}, x_s^{(1)}), \cdots, (x_o^{(N)}, x_s^{(N)})\}$, *Magical* treats $x_s^{(i)}$ as the positive sample for $x_o^{(i)}$, and $x_s^{(j)}$ $(j \neq i)$ as a negative sample. Contrastive learning aligns the representations by minimizing the distance between $x^{(i)}$ and $x_+^{(i)}$, while maximizing the distance between $x^{(i)}$ and $x_-^{(i)}$. *Magical* denotes the set of positive samples for $x$ as $\chi^+$, and the set of negative samples as $\chi^-$, then the training objective can be formally defined as:

$$\mathcal{L}_{contra}(x, \chi^+, \chi^-) = -\log \frac{\sum_{x' \in \chi^+} \exp(sim(x, x^{'}/\tau)}{\sum_{x' \in (\chi^+, \chi^-)} \exp(sim(x, x^{'})/\tau)}. \tag{3}$$

where $sim(\cdot)$ refers to cosine similarity between representations, and $\tau$ is the temperature.

Regarding implementation details, since *Magical* receives only expert language as input in the MLLG task, we construct a *cached key dictionary* to leverage lay language as contrastive targets. Specifically, during each training iteration, we use the matrix $A$ retained from the previous iteration to encode the lay language into representations, which are then stored as a key dictionary and cached for use in the current iteration's `Semantic Contrastive Learning`.

### 3.3 Layperson-tailored Adaptation on $B$

**Router-Controlled *vs.* Switch-Controlled.** *Magical* explores two mechanisms for branch control variable $\alpha_i$ in Eq. 2 to aggregate matrices $B$: *Router-Controlled Selection* and *Switch-Controlled Selection*. In this research, *Magical* defines them as follows:

- **Router-Controlled Selection**: Router-Controlled Selection defines $\alpha$ as a continuous probability distribution, leveraging a *Soft Selection* mechanism to aggregate the performance of multiple $B$ matrices. Specifically, this can be formulated as $\sum_{i=1}^{N} \alpha_i = 1$. Typically, multiple matrices $B$ are activated, and the LLM integrates their outputs to handle complex tasks.
- **Switch-Controlled Selection**: Switch-Controlled Selection defines $\alpha$ as a one-hot vector, employing a *Hard Selection* mechanism to choose a specific branch. Specifically, this entails that $\alpha_i = 1$ while $\alpha_j = 0$ for all $i \neq j$. In this setting, the LLM typically does not autonomously identify the task. Only a single matrix $B$ is activated, which reduces interference from other $B$ matrices.

**Recommendation-guided Switch.** Unlike existing mainstream approaches [28, 35, 36] that employ *Router-Controlled Selection* to enable LoRA for multi-task learning, *Magical* adopts *Switch-Controlled Selection* for MLLG task to mitigate the interference caused by the heterogeneity of multi-source lay language dataset. This design is motivated by the observation that, in typical multi-task learning frameworks, the differences between tasks are often easily distinguishable, allowing simple prefix prompts [37, 38] and the in-context learning ability [39] of LLMs to effectively combine multiple $B$ matrices. However, in the heterogeneous data setting of the MLLG task, such simple

Table 2: Performance comparison of Prompt, various LoRA variants, and *Magical* across three backbone LLMs on three MLLG datasets. All evaluation metrics are defined such that higher values indicate better performance (↑). The best results are highlighted in **bold**, while the second-best results are underlined. The *Impro* (%) indicates the relative improvement of *Magical* over the second-best performance.

| Methods | #Params | Cochrane | | | | eLife | | | | Plos_genetics | | | |
|---|---|---|---|---|---|---|---|---|---|---|---|---|---|
| | | R-1 | R-2 | R-L | BLEU | R-1 | R-2 | R-L | BLEU | R-1 | R-2 | R-L | BLEU |
| *LLaMA3.2-3B-Instruct* | | | | | | | | | | | | | |
| Prompt | N/A | 39.81 | 11.37 | 36.51 | 5.28 | 37.82 | 8.22 | 35.47 | 3.08 | 37.88 | 7.78 | 35.08 | 4.13 |
| LoRA [12] | 36M | 40.01 | 16.33 | 37.71 | 10.37 | 46.69 | 13.48 | 44.57 | 7.36 | 44.92 | 13.62 | 41.42 | 8.80 |
| rsLoRA [42] | 36M | 40.85 | 16.38 | 38.44 | 10.23 | 45.41 | 13.02 | 43.30 | 7.09 | 43.75 | 13.91 | 40.54 | 7.79 |
| DoRA [43] | 37M | 40.03 | 16.28 | 37.81 | 10.26 | 42.93 | 12.01 | 40.84 | 6.16 | 41.17 | 12.81 | 38.43 | 6.44 |
| PiSSA [44] | 36M | 39.75 | 16.14 | 37.38 | 10.52 | 42.89 | 11.99 | 40.85 | 6.26 | 43.51 | 13.91 | 40.45 | 7.79 |
| *Magical* | 24M | **45.33** | **19.39** | **42.36** | **16.66** | **49.16** | **14.68** | **46.91** | **8.30** | **47.50** | **15.47** | **44.03** | **10.24** |
| *Impro* (%) | 33.33 | 10.97 | 18.38 | 10.20 | 58.37 | 5.29 | 8.90 | 5.25 | 12.77 | 5.74 | 11.21 | 6.30 | 16.36 |
| *LLaMA3.1-8B-Instruct* | | | | | | | | | | | | | |
| Prompt | N/A | 41.67 | 12.50 | 38.60 | 6.23 | 35.82 | 9.44 | 33.36 | 1.85 | 39.13 | 8.67 | 36.16 | 4.66 |
| LoRA [12] | 62M | 40.19 | 16.56 | 37.53 | 12.06 | 49.40 | 14.88 | 47.29 | 8.53 | 47.55 | 16.34 | 43.98 | 11.54 |
| rsLoRA [42] | 62M | 43.30 | 18.12 | 40.62 | 12.89 | 49.33 | 15.01 | 47.25 | **8.71** | 42.19 | 12.86 | 39.36 | 6.87 |
| DoRA [43] | 64M | 43.24 | 18.28 | 40.60 | 13.50 | 48.47 | 14.67 | 46.41 | 8.44 | 42.48 | 13.06 | 39.59 | 7.17 |
| PiSSA [44] | 62M | 42.95 | 17.80 | 40.23 | 13.89 | 48.83 | 14.66 | 46.74 | 8.40 | 39.64 | 11.62 | 37.26 | 5.62 |
| *Magical* | 42M | **45.71** | **19.52** | **42.79** | **16.68** | **50.44** | **15.49** | **48.02** | 8.67 | **48.77** | **16.64** | **45.06** | **11.86** |
| *Impro* (%) | 32.23 | 5.57 | 6.78 | 5.34 | 20.09 | 2.11 | 3.20 | 1.54 | -0.46 | 2.57 | 1.84 | 2.46 | 2.77 |
| *Qwen2.5-7B-Instruct* | | | | | | | | | | | | | |
| Prompt | N/A | 44.53 | 15.57 | 41.52 | 11.42 | 35.09 | 8.85 | 32.29 | 1.80 | 44.93 | 12.50 | 41.14 | 7.73 |
| LoRA [12] | 60M | 44.23 | 19.37 | 41.60 | 14.50 | 47.38 | 13.90 | 45.42 | 7.93 | 46.41 | 16.29 | 42.95 | 9.94 |
| rsLoRA [42] | 60M | 45.22 | 19.48 | 42.46 | 15.06 | 48.70 | 14.54 | 46.48 | 7.76 | 46.26 | 15.60 | 42.87 | 9.90 |
| DoRA [43] | 61M | 45.65 | 19.64 | 42.87 | 15.45 | 47.38 | 13.90 | 45.24 | 7.70 | 46.64 | 15.97 | 43.24 | 10.12 |
| PiSSA [44] | 60M | 44.87 | 19.30 | 42.12 | 14.52 | 48.86 | 14.46 | 46.68 | 7.82 | 46.81 | 15.93 | 43.57 | 10.48 |
| *Magical* | 42M | **47.42** | **20.81** | **44.38** | **17.89** | **50.50** | **15.28** | **48.16** | **8.66** | **48.54** | **16.39** | **44.79** | **11.42** |
| *Impro* (%) | 30.00 | 3.88 | 5.96 | 3.52 | 15.79 | 3.36 | 5.09 | 3.17 | 9.21 | 3.70 | 0.61 | 2.80 | 8.97 |

prompts become ineffective. This is because overly simplistic instructions fail to comprehensively articulate the target lay-style, and demonstrations [40] may significantly increase sequence length, thereby raising the risk of lost-in-the-middle [41]. In subsequent experiments, we validated the dilemma faced by *Router-Controlled Selection* in the MLLG task and demonstrated the rationality of *Switch-Controlled Selection*.

Due to the limitations of LLMs in autonomously selecting appropriate lay-style, *Magical* adopts a divide-and-conquer principle [23] by employing an external *Recommendation Agent* to handle lay-style selection. It further introduces an open *Interface* to bridge the *Switch* mechanism with the *Recommendation Agent*, enabling dynamic switching to the most suitable matrix $B$ for the target lay-style generation. The core insight of *Magical* lies in avoiding the excessive accumulation of redundant optimization objectives within a single low-rank subspace, which would otherwise compromise the quality of lay language generation.

## 4 Experiment

### 4.1 Experimental Setups

**Network Architecture, Datasets and Metrics.** Our experiments were based on various backbone LLMs, including LLaMA3 series [45] of various sizes (LLaMA3.2-3B-Instruct and LLaMA3.1-8B-Instruct) and Qwen2.5-7B-Instruct [46]. To evaluate the effectiveness of *Magical*, we conducted experiments on three publicly available real-world MLLG datasets, including Cochrane [24], eLife [25], and Plos_genetics [2]. We followed the official data splits to construct the training and test sets. Detailed statistics of these datasets are provided in Appendix. A. Following the evaluation protocol of existing literature [9], we adopted *ROUGE-1* (R-1), *ROUGE-2* (R-2), *ROUGE-L* (R-L), and *BLEU* as evaluation metrics for the MLLG task, with detailed descriptions available in Appendix. B.

**Baselines.** We adopt the following state-of-the-art approaches as our compared baselines.

Table 3: Ablation Studies of *Magical* using LLaMA3.1-8B-Instruct as backbone LLM on three MLLG datasets. **SRLI**: Semantic-Relevant Layer Identification. **SCL**: Semantic Contrastive Learning. All evaluation metrics are defined such that higher values indicate better performance (↑).

| Methods | Cochrane | | | | eLife | | | | Plos_genetics | | | |
| --- | --- | --- | --- | --- | --- | --- | --- | --- | --- | --- | --- | --- |
| | R-1 | R-2 | R-L | BLEU | R-1 | R-2 | R-L | BLEU | R-1 | R-2 | R-L | BLEU |
| *LLaMA3.1-8B-Instruct* | | | | | | | | | | | | |
| *Magical* | 45.71 | 19.52 | 42.79 | 16.68 | 50.44 | 15.49 | 48.02 | 8.67 | 48.77 | 16.64 | 45.06 | 11.86 |
| w/o SRLI | 41.41 | 17.22 | 38.65 | 12.20 | 49.83 | 15.04 | 47.60 | 8.59 | 47.97 | 16.33 | 44.31 | 11.57 |
| w/o SCL | 45.09 | 19.31 | 42.26 | 14.79 | 49.67 | 14.96 | 47.41 | 8.30 | 48.35 | 16.35 | 44.74 | 11.62 |
| → Single $B$ | 41.32 | 17.45 | 38.72 | 11.84 | 48.25 | 14.59 | 46.27 | 8.49 | 41.98 | 12.85 | 39.17 | 6.79 |
| Switch→Router | 41.77 | 16.93 | 39.34 | 11.21 | 47.76 | 14.08 | 45.64 | 7.86 | 41.01 | 12.39 | 38.27 | 6.19 |

- **Prompt**: Prompt leverages the in-context learning [39] ability to achieve MLLG. Specifically, we use a well-crafted prompt to describe the target lay-style with demonstrations. (See Appendix. D for detailed prompts).

- **Various LoRA Variants**: We adopt LoRA [12] (multi-LoRA version) and several recently proposed LoRA variants as our baseline models, including HydraLoRA [28], rsloRA [42], DoRA [43], and PiSSA [44]. It is worth noting that HydraLoRA [28] serves as an inspiration for *Magical*, and is therefore treated as a variant of *Magical* in our ablation studies.

**Implement Details.** We used the PyTorch library to implement all the algorithms based on the open-source HuggingFace transformers [47]. The experiments were conducted on 8 NVIDIA-H20-96GB GPUs. For each experimental setup, we trained all LLMs for 5 epoch with DeepSpeed ZeRO 2 Offload [48]. We utilized the AdamW optimizer and a cosine learning rate scheduler, with a warm-up ratio set to 0.1. For the *Recommendation Agent* in `Layperson-tailored Adaptation`, we adopt a manually specified matrix $B$, corresponding to a *Recommendation Agent* with 100% recommendation accuracy.

## 4.2 Experimental Results

**Comparison with Recent Literature.** As shown in Table. 2, we compare *Magical* with Prompt method, vanilla LoRA, and various LoRA variants across three MLLG tasks. To comprehensively evaluate the effectiveness of *Magical*, we employ backbone LLMs with diverse architectures and scales. The experimental results demonstrate that *Magical* achieves the best performance in most cases, ranking second only to rsLoRA on the BLEU for the eLife dataset. Moreover, compared to the second-best performance, *Magical* yields an average relative improvement of 4.80%↑ in R-1, 6.89%↑ in R-2, 4.51%↑ in R-L and 15.99%↑ in BLEU across all settings, confirming its strong performance. More importantly, while achieving performance improvements, *Magical* also introduces average 31.66%↓ trainable parameters. Notably, the Prompt method consistently performs the worst in the majority of scenarios, especially on the BLEU metric. This supports our earlier claim that simple prompts are insufficient to effectively capture the target lay-style, making it challenging to generate language tailored for laypersons.

**Ablation Studies.** We verified the effectiveness of each module by removing some modules from *Magical* and evaluated the modified models using the LLaMA3.1-8B-Instruct as backbone LLM across three MLLG datasets. The experimental results were presented in Table. 3. The experimental results provide deeper insights into the design of *Magical*. We first analyze the modules added to matrix $A$. When the `Semantic-Relevant Layer Identification (SRLI)` component is removed—i.e., `Semantic Invariance Constraint` are imposed across all Transformer layers of the LLM—there is a significant performance drop (e.g., an average 1.90%↓ R-1). This suggests that not all Transformer layers are responsible for semantic activation, and indiscriminately applying constraints may harm other functionalities. Similarly, removing the `Semantic Contrastive Learning (SCL)` module also leads to a noticeable performance degradation (e.g., an average 0.83%↓ BLEU). This indicates that without the optimization of a latent objective promoting semantic invariance, the low-rank projection of LoRA may introduce semantic shifts, thereby undermining model performance.

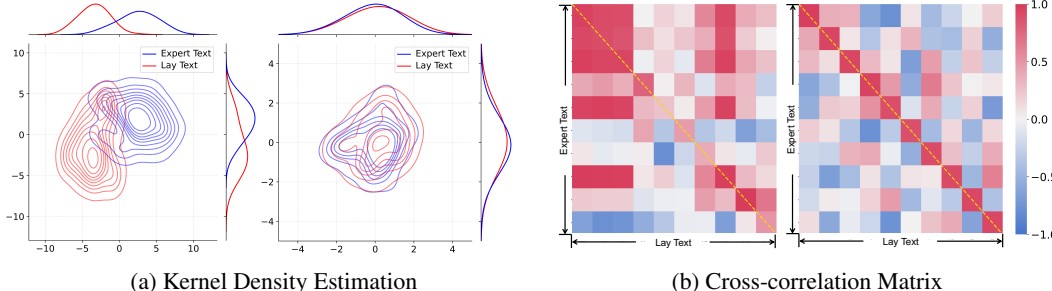

(a) Kernel Density Estimation          (b) Cross-correlation Matrix

Figure 4: (a) shows Kernel Density Estimation based on the projections of activations obtained with / without the application of `Semantic Invariance Constraint` on $A$. (b) presents Cross-correlation Matrix for the representations of expert text and lay text under matrix $A$ with / without the use of `Semantic Invariance Constraint` on $A$.

We further investigate the design choices regarding matrix $B$ in *Magical*. Replacing multiple $B$ matrices with a single matrix $B$ results in a clear performance drop (e.g., an average 4.46%↓ R-1). This is because a single matrix $B$ struggles to project expert texts into diverse lay-style texts effectively. Moreover, when the *Switch-Controlled Selection* mechanism is replaced with a *Router-Controlled* one, the performance becomes comparable to using only a single $B$ matrix. This suggests that the routing mechanism does not effectively guide the LLM in selecting an appropriate $B$, thus failing to exploit the advantages brought by multiple $B$ for layperson-tailored adaptation.

**Verification of Semantic Invariance.** As shown in Fig. 4a, we randomly selected a MLP layer with / without the `Semantic Invariance Constraint` applied on matrix $A$, and projected the representations of expert texts and generated lay texts onto the first two singular directions of the semantic-relevant subspace. We then plotted their respective kernel density estimation distributions. The experimental results demonstrate that applying the `Semantic Invariance Constraint` on $A$ effectively suppresses semantic shifts during the low-rank projection process. Furthermore, we randomly selected a matrix $A$ and visualized the cross-correlation matrix between the representations of expert texts and their corresponding lay texts after transformation by $A$ as shown in Fig. 4b. The results indicate that this matrix $A$ is capable of mapping expert texts and lay texts into a shared representation space, thereby preserving semantic fidelity in the generated lay text. The case study presented in Appendix. E.10 provides a more detailed illustration of the semantic degradation introduced by LoRA and the semantic fidelity preserved by *Magical*.

**Extended Investigations and Key Insights.** We conducted a more detailed analysis of `Layperson-tailored Adaptation`, including an investigation into the **Dilemma of Router-Controlled Selection**, and demonstrated that *Router-Controlled Selection completely fails in scenarios where the LLM autonomously selects matrices $B$*. Furthermore, we analyzed the **Sensitivity of Recommendation Performance** and observed that, *although the performance of Magical degrades with declining recommendation system quality, it still consistently outperforms standard baseline models*. We also examined the **Sensitivity of Rank**, and showed that *Magical continues to surpass baseline algorithms even when different ranks are used*. Detailed experimental results can be found in Appendices. E.1, E.2, and E.3. In addition, we provide **Case Study** in Appendix. E.10 to further analyze the superiority of *Magical*.

## 5 Limitation, Future Works and Conclusion

Despite the promising results obtained in our work, it is important to acknowledge the limitations. Considering that *Magical* employs a divide-and-conquer strategy, which relies on an external *Recommendation Agent* to select suitable layperson styles. Although effective—and empirically shown to outperform baselines even with a suboptimal recommender—the *Recommendation Agent* itself was not implemented in this work (see Implement Details 4.1) due to the lack of corresponding user data. Recent advances [49, 50] in recommendation systems suggest that building such agent requires modeling the target users based on their individual profiles and behavioral histories. However, the current MLLG dataset does not provide these layperson-specific information. We also notice that this same linguistic complexity also poses challenges for interpreting the outputs of electronic health

record (EHR) predictive modeling [51–53]. In future work, adapting *Magical* to translate these technical outputs into understandable narratives could significantly improve patient engagement and shared decision-making. We will also focus on the development of lay-style recommendation agents under the cold-start setting [54]. Existing relevant strategies include meta-learning [55, 56], pre-trained models [57, 58], and reinforcement learning [59, 60], among others. Integrating these recent advancements into our lay-style recommendation agents could more effectively support *Magical* in achieving lay-style recommendation and lay-style generation in real-world scenarios.

In this work, we conducted an in-depth investigation into the limitations of using LoRA to fine-tune LLMs for the MLLG task, particularly its difficulty in preserving semantic fidelity and adapting to diverse lay-style generation patterns introduced by heterogeneous data. To address these challenges, we proposed *Magical*, an asymmetric LoRA-based framework that incorporates a `Semantic Invariance Constraint` and `Layperson-tailored Adaptation` to enhance both semantic alignment and stylistic flexibility. While *Magical* does not yet integrate a dedicated *Recommendation Agent* for lay-style selection in this work, we advocate a novel perspective: decoupling lay-style recommendation from lay-style generation. We envision a multi-agent system where separate, specialized modules handle generation and recommendation respectively, enabling mutual reinforcement between the two components. This collaborative design offers a promising direction for advancing research in the MLLG domain.

# 6 Acknowledgement

This work was supported by National Natural Science Foundation of China (62402017), Beijing Traditional Chinese Medicine Science and Technology Development Fund (BJZYZD-2025-13), Peking University Clinical Medicine Plus X (Young Scholars Project-the Fundamental Research Funds for the Central Universities PKU2025PKULCXQ024; Pilot Program-Key Technologies Project 2024YXXL-HGG007), and Peking University "TengYun" Clinical Research Program (TY2025015). Liantao Ma was supported by Beijing Natural Science Foundation (L244063, L244025), Beijing Municipal Health Commission Research Ward Excellence Clinical Research Program (BRWEP2024W032150205), and Xuzhou Scientific Technological Projects (KC23143). Junyi Gao acknowledges the receipt of studentship awards from the Health Data Research UK-The Alan Turing Institute Wellcome PhD Programme in Health Data Science (grant 218529/Z/19/Z) and Baidu Scholarship.

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

# A  Statistic of Datasets

We present the detailed size and composition of three MLLG datasets in Table. 4.

Table 4: Statistical information of three MLLG datasets.

| Datasets | #Train | #Test |
|---|---|---|
| Cochrane [2] | 3,979 | 480 |
| eLife [25] | 4,587 | 241 |
| Plos_genetics [2] | 4,000 | 300 |

# B  Details of Metrics

## B.1  Heterogeneity Assessment Metrics

**Word Count**   Word Count a quantitative analysis of sentence length, operationalized as the average number of words per sentence. This metric serves as an important stylistic feature in textual analysis.

The average Word Count was calculated using the following formula:

$$\text{Word Count} = \frac{\text{Total number of words}}{\text{Total number of sentences}} \tag{4}$$

**Document Complexity and Readability Score (DCRS)**   The Document Complexity and Readability Score (DCRS) represents a significant advancement in the field of text readability assessment, offering a multidimensional approach to quantifying textual complexity. Unlike traditional readability formulas such as Flesch-Kincaid or SMOG that primarily rely on surface-level features, DCRS integrates lexical, syntactic, semantic, and discourse-level features to provide a more comprehensive evaluation of document readability.

DCRS is defined as a weighted composite metric:

$$\text{DCRS} = \alpha L + \beta S + \gamma C + \delta D \tag{5}$$

where $L$ represents lexical complexity, $S$ denotes syntactic complexity, $C$ indicates conceptual density, and $D$ measures discourse coherence. The coefficients $\alpha$, $\beta$, $\gamma$, and $\delta$ are empirically derived weights that reflect the relative contribution of each component to overall readability, with $\alpha + \beta + \gamma + \delta = 1$.

**Readability Evaluation of DeepSeek-V3**   We use DeepSeek-V3 to evaluate the readability of the text, employing the following prompt:

---

**Prompt used for generating readability evaluation by DeepSeek-V3**

```
TEMPLETE = r"""[Task]
Evaluate the readability of two medical texts (1-10 scale) using
these criteria:
1.  Conciseness (simple vocabulary/short sentences)
2.  Structural clarity (logical flow/hierarchical explanations)
3.  Terminology handling (necessary jargon with explanations)
4.  Comprehension ease (quick understanding for non-experts)

[The Start of Medical Text 1]
{medical_text_1}
[The End of Medical Text 1]
```

```
 [The Start of Medical Text 2]
{medical_text_2}
 [The End of Medical Text 2]

 [Scoring Guide]
10:  Fully understandable without medical background
7-9:  Minimal jargon explained through context
4-6:  Requires basic medical knowledge
1-3:  Excessive unexplained technical terms

 [Output Instructions]
- Output ONLY two numerical scores (e.g., "7 9") on the first line
- Do NOT include any explanations, analysis, or additional text"""
```

## B.2 Content Preservation and Quality Metrics

**ROUGE Metrics**  The Recall-Oriented Understudy for Gisting Evaluation (ROUGE) suite provides metrics for assessing the quality of generated summaries by measuring overlap with reference texts:

- *ROUGE-1* (R-1) calculates the overlap of unigrams (individual words) between the generated text and reference text, reflecting content coverage at the word level.

- *ROUGE-2* (R-2) measures the overlap of bigrams (word pairs) between the generated and reference texts, capturing local word order and phrasal information.

- *ROUGE-L* (R-L) computes the longest common subsequence between the generated and reference texts, evaluating fluency and word order without requiring consecutive matches.

These ROUGE variants collectively provide insights into how well the simplified text preserves essential content from the expert source while using accessible language. Higher ROUGE scores indicate better content preservation.

**BLEU Score**  The Bilingual Evaluation Understudy (BLEU) metric, initially developed for machine translation evaluation, has been adapted for text simplification tasks. BLEU calculates the precision of n-gram matches between the generated text and reference text, with a penalty for brevity. The score ranges from 0 to 1, with higher values indicating greater similarity to reference texts.

BLEU is particularly valuable for assessing the linguistic quality and fluency of generated lay texts, complementing the recall-oriented nature of ROUGE metrics. The formula incorporates various n-gram precisions:

$$\text{BLEU} = \text{BP} \cdot \exp\left(\sum_{n=1}^{N} w_n \log p_n\right) \tag{6}$$

where BP is the brevity penalty, $w_n$ are weights for different n-gram precisions, and $p_n$ represents the precision for n-grams of length $n$.

## C  Hyperparameter Settings

We present the detailed hyperparameter settings in Table. 5.

Table 5: Hyperparameter settings for fine-tuning the LLM with *Magical*. Contrastive Loss Weight denotes the weight of the contrastive learning loss (Eq. 3) in the composite loss. $\tau$ refers to the temperature coefficient used in the contrastive loss (Eq. 3). $K$ indicates the number of top-$K$ Transformer layers selected in the `Semantic-Relevant Layer Identification` module.

| Hyperparameter | LLaMA3.2-3B-Instruct | LLaMA3.1-8B-Instruct | Qwen2.5-7B-Instruct |
|---|---|---|---|
| Batch Size | 64 | 64 | 128 |
| Learning Rate | 1e-4 | 3e-4 | 3e-4 |
| Contrastive Loss Weight | 0.5 | 0.3 | 0.5 |
| $\tau$ | 0.5 | 0.5 | 0.5 |
| $K$ | 32 | 16 | 16 |

## D   Prompt Method

---

**Prompt used for baseline model to finish MLLG**

```
Simplify the following medical text for general public understanding
using these criteria:

1.  Conciseness (simple vocabulary/short sentences)
2.  Structural clarity (logical flow/hierarchical explanations)
3.  Terminology handling (necessary jargon with explanations)
4.  Comprehension ease (quick understanding for non-experts)

Below is an example:
Medical Text:  {medical_text}
Simplified Text:  {simplified_text}

Now please perform the simplification:
Medical Text:  {medical_text_1}
```

---

## E   Additional Experiments

### E.1   Dilemma of Router-Controlled Selection

We further conduct an in-depth analysis of the *Router-Controlled Selection*. Specifically, we randomly select a router and visualize the selection results for the target lay-style. The corresponding confusion matrix is presented in Fig. 5. Experimental results indicate that the router fails to correctly identify the target lay-style, as evidenced by the confusion matrix showing near-random selection behavior. This is primarily due to the insufficiency of expert text and limited prompts in accurately representing the target lay-style. In particular, for the MLLG task, lay-styles exhibit substantial differences in aspects such as depth of simplification and simplification strategies, making it difficult for the LLM to perform correct routing based on such vague input. These findings highlight the effectiveness of `Recommendation-guided Switch` proposed by *Magical*.

### E.2   Sensitivity of Recommendation Performance

In this work, we employ a manually selected approach to simulate the behavior of a *Recommendation Agent* that has not yet been fully implemented in this work. This implies that the *Recommendation Agent* operates with an assumed 100% accuracy, which is rarely achievable in real-world scenarios. Therefore, it is important to investigate the sensitivity of *Magical* to the accuracy of recommendations. Specifically, we manually introduce incorrect recommendations into randomly selected cases to simulate a more realistic setting with imperfect *Recommendation Agents*. As shown in Fig. 6, we systematically reduce the recommendation accuracy to 95%, 90%, 80%, 70%, and 60%, and evaluate the corresponding performance of *Magical*. Experimental results show that the performance of *Magical* degrades as the recommendation accuracy decreases. However, it is noteworthy that even

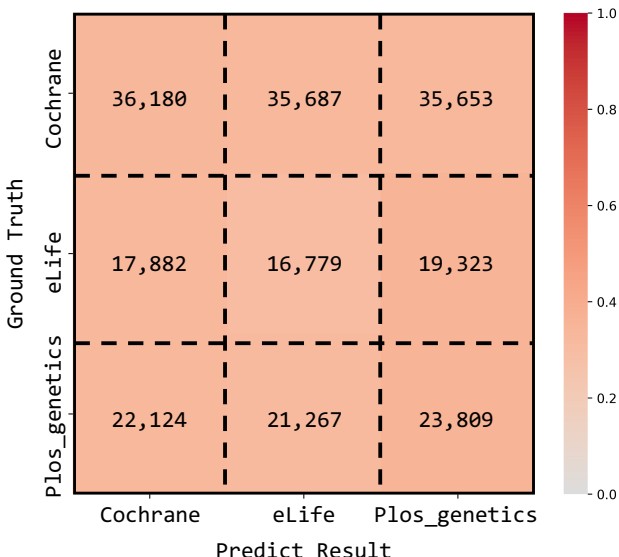

Figure 5: Confusion matrix of predictions made by *Router* selection matrix $B$ for target lay-style.

with only 60% recommendation accuracy, *Magical* still consistently outperforms naive LoRA-based methods. This demonstrates that the superiority of *Magical* is not solely due to external manual selection of target styles, but also stems from its other advanced components, such as the `Semantic Invariance Constraint`.

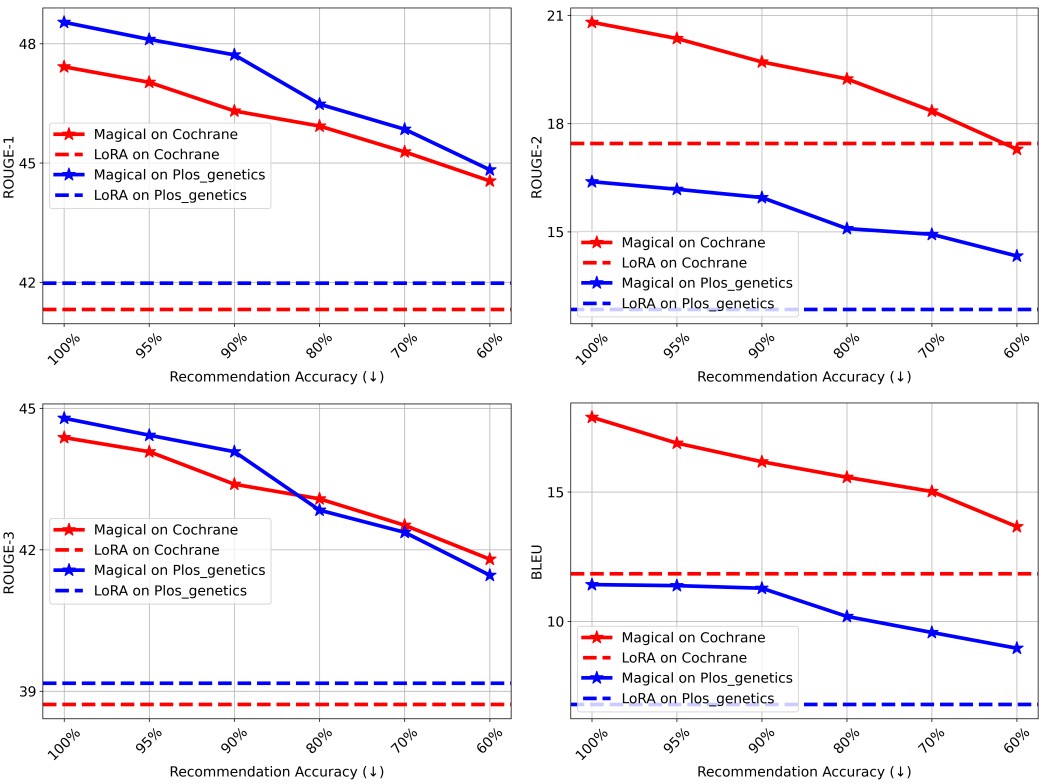

Figure 6: Sensitivity analysis of accuracy of *Recommendation Agent* in *Magical*.

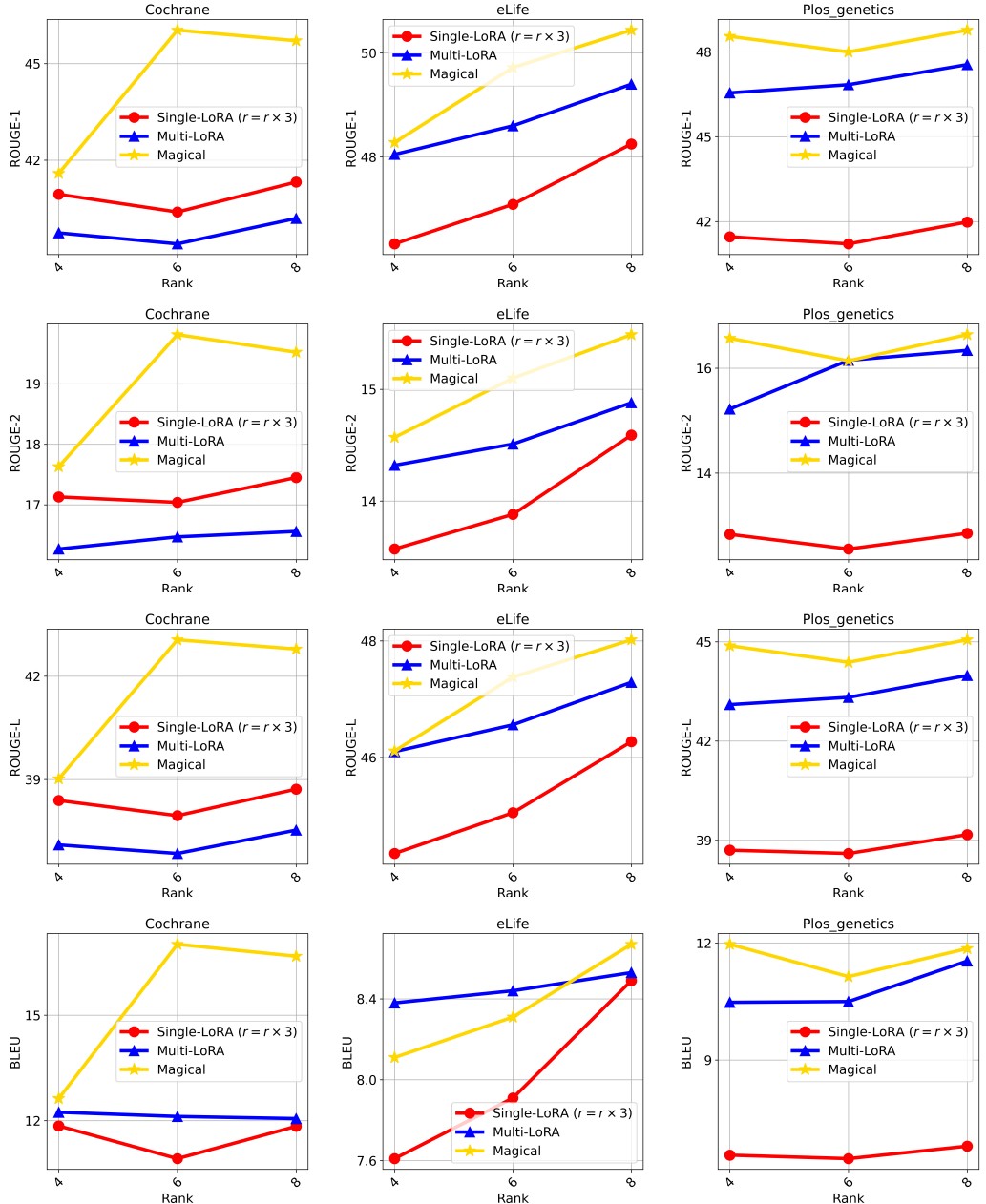

Figure 7: Sensitivity analysis of Rank in Single-LoRA, Multi-LoRA and *Magical*. Please note that the rank used in Multi-LoRA is consistent with that of *Magical*. In order to ensure parameter equivalence during training, Single-LoRA adopts a higher rank than Multi-LoRA—specifically, its rank is $3\times$ that of both Multi-LoRA and *Magical*.

## E.3 Sensitivity of Rank

We conduct a sensitivity analysis on the rank of *Magical* and compare it with Multi-LoRA of the same rank and Single-LoRA with a rank $3\times$ larger. The experimental results are shown in Fig. 7. As illustrated, *Magical* consistently outperforms both Single-LoRA and Multi-LoRA in the vast majority of cases.

### E.4 Evaluation of the Actual Quality of Generated Lay Texts

In this experiment, we introduce two complementary evaluation strategies to reflect the actual perceived quality of generated lay texts.

**Large Language Models as Evaluators (Full-scale Evaluation).** We used o3-mini and DeepSeek-R1 to rate outputs from *Magical* and all baseline systems on a 1–10 scale. These models acted as automatic evaluators simulating expert judgment across the full test sets. In this experiment, we selected LLaMA3.1-8B-Instruct as backbone LLM. As shown in Table. 6, *Magical* consistently outperformed all baselines across all datasets, indicating its robust ability to generate high-quality lay texts.

Table 6: Performance comparison of various LoRA variants and *Magical* using LLMs as evaluator. All evaluation metrics are defined such that higher values indicate better performance (↑). The best results are highlighted in **bold**.

| Methods | Cochrane | | eLife | | Plos_genetics | |
|---------|----------|------------|----------|------------|----------|------------|
| | o3-mini | DeepSeek-R1 | o3-mini | DeepSeek-R1 | o3-mini | DeepSeek-R1 |
| *LLaMA3.1-8B-Instruct* | | | | | | |
| LoRA | 6.57 | 6.44 | 6.94 | 7.02 | 7.35 | 7.43 |
| rsLoRA | 6.75 | 6.43 | 7.28 | 7.58 | 7.07 | 6.99 |
| DoRA | 6.77 | 6.56 | 7.19 | 7.66 | 7.11 | 7.28 |
| PiSSA | 6.54 | 6.33 | 7.30 | 7.36 | 6.83 | 6.87 |
| *Magical* | **7.53** | **7.66** | **7.66** | **7.92** | **8.02** | **8.01** |

**Human Expert Evaluation (Focused-scale Evaluation).** To further validate *Magical*'s quality in a real-world setting, we randomly sampled 20 examples from each MLLG dataset (60 samples total), and invited two medical data analysis experts to finish the human evaluation. Each expert was shown the expert-level source text, and two lay versions generated by LoRA and *Magical*—with the generation sources anonymized to ensure fairness. The results, presented in Table.7, show that both experts consistently rated *Magical* higher than LoRA, validating that *Magical* substantially improves the clarity and quality of generated lay texts.

Table 7: Performance comparison of LoRA and *Magical* using Expert as evaluator. All evaluation metrics are defined such that higher values indicate better performance (↑). The better results are highlighted in **bold**.

| Methods | Cochrane (#20) | | eLife (#20) | | Plos_genetics (#20) | |
|---------|----------|----------|----------|----------|----------|----------|
| | expert-1 | expert-2 | expert-1 | expert-2 | expert-1 | expert-2 |
| *LLaMA3.1-8B-Instruct* | | | | | | |
| LoRA | 6.90 | 7.10 | 6.95 | 6.90 | 7.05 | 7.05 |
| *Magical* | **8.55** | **8.35** | **7.85** | **7.40** | **8.60** | **8.50** |

### E.5 Semantic Invariance Evaluation via Quantitative Metric

In this experiment, we have incorporated BERTScore as a complementary evaluation metric. We select LLaMA3.1-8B-Instruct as the backbone LLM and conduct our experiments accordingly. The new results are presented in Table. 8, and they align well with the trends we observed using KDE, thereby further confirming the semantic fidelity of *Magical*.

### E.6 Compared to Full-Parameter Fine-Tuning (FFT)

We added Full-Parameter Fine-Tuning (FFT) as a new baseline in our experiments. As shown in Table. 9, *Magical* consistently outperforms the FFT across all datasets and three different backbone

Table 8: Semantic invariance evaluation of Prompt, various LoRA variants, and *Magical* using BertScore on three MLLG datasets. BertScore is defined such that higher values indicate better performance (↑). The best results are highlighted in **bold**.

| | Cochrane | eLife | Plos_genetics |
|---|---|---|---|
| *LLaMA3.1-8B-Instruct* | | | |
| Prompt | 85.81 | 85.25 | 86.73 |
| LoRA | 88.22 | 84.50 | 89.61 |
| rsLoRA | 87.73 | 84.71 | 88.71 |
| DoRA | 88.77 | 84.71 | 89.06 |
| PiSSA | 89.03 | 84.83 | 90.01 |
| *Magical* | **90.79** | **86.80** | **91.94** |

Table 9: Performance comparison of FFT and *Magical* across three backbone LLMs on three MLLG datasets. All evaluation metrics are defined such that higher values indicate better performance (↑). The better results are highlighted in **bold**.

| Methods | #Params | Cochrane | | | | eLife | | | | Plos_genetics | | | |
|---|---|---|---|---|---|---|---|---|---|---|---|---|---|
| | | R-1 | R-2 | R-L | BLEU | R-1 | R-2 | R-L | BLEU | R-1 | R-2 | R-L | BLEU |
| *LLaMA3.2-3B-Instruct* | | | | | | | | | | | | | |
| FFT | 3,606M | 44.19 | 18.28 | 41.37 | 13.84 | 47.01 | 12.72 | 45.01 | 7.20 | 43.50 | 14.12 | 40.25 | 7.79 |
| *Magical* | 24M | **45.33** | **19.39** | **42.36** | **16.66** | **49.16** | **14.68** | **46.91** | **8.30** | **47.50** | **15.47** | **44.03** | **10.24** |
| *LLaMA3.1-8B-Instruct* | | | | | | | | | | | | | |
| FFT | 8,030M | 43.2 | 18.34 | 40.27 | 15.90 | 48.76 | 13.30 | 46.92 | 7.82 | 44.81 | 13.99 | 41.61 | 7.99 |
| *Magical* | 42M | **45.71** | **19.52** | **42.79** | **16.68** | **50.44** | **15.49** | **48.02** | **8.67** | **48.77** | **16.64** | **45.06** | **11.86** |
| *Qwen2.5-7B-Instruct* | | | | | | | | | | | | | |
| FFT | 7,615M | 45.99 | 19.62 | 42.58 | 15.81 | 47.73 | 14.49 | 45.36 | 7.20 | 45.49 | 15.64 | 43.20 | 9.97 |
| *Magical* | 42M | **47.42** | **20.81** | **44.38** | **17.89** | **50.50** | **15.28** | **48.16** | **8.66** | **48.54** | **16.39** | **44.79** | **11.42** |

LLMs. Moreover, FFT introduces over $150\times$ more trainable parameters compared to *Magical*. These results highlight *Magical*'s dual advantages in performance and parameter efficiency.

### E.7 Compared to Large-Scale LLMs

We added o3-mini as a new baseline and applied the prompt-based method described in Section. 4.1. As shown in Table. 10, the performance of o3-mini with prompting was suboptimal. We argue that simple prompt strategies are insufficient to comprehensively capture the target lay-style, making non-learning-based approaches poorly adaptable to the target dataset.

Table 10: Performance comparison of o3-mini and *Magical* on three MLLG datasets. All evaluation metrics are defined such that higher values indicate better performance (↑). The better results are highlighted in **bold**.

| Methods | Cochrane | | | | eLife | | | | Plos_genetics | | | |
|---|---|---|---|---|---|---|---|---|---|---|---|---|
| | R-1 | R-2 | R-L | BLEU | R-1 | R-2 | R-L | BLEU | R-1 | R-2 | R-L | BLEU |
| o3-mini | 46.50 | 14.30 | 42.82 | 9.00 | 38.15 | 9.48 | 34.75 | 2.46 | 42.59 | 10.42 | 38.76 | 5.65 |
| *Magical* | **47.42** | **20.81** | **44.38** | **17.89** | **50.50** | **15.28** | **48.16** | **8.66** | **48.54** | **16.39** | **44.79** | **11.42** |

### E.8 Sensitivity Analysis of $K$

we have added a sensitivity analysis of the top-$K$ selection based on LLaMA3.1-8B-Instruct. The experimental results, shown in Table. 11, reveal that both overly small values of $K$ (e.g., $K = 8$) and overly large values (e.g., $K = 24$ or 32) result in degraded performance. This is because selecting too many layers introduces semantic constraints to layers that are not semantically relevant, leading

to interference with the LLM's functional integrity. Conversely, too few layers may fail to impose constraints on all semantically relevant layers, resulting in semantic shift.

Table 11: Sensitivity Analysis of $K$ in *Magical* on three MLLG datasets. All evaluation metrics are defined such that higher values indicate better performance (↑). The best results are highlighted in **bold**.

| Methods | Cochrane | | | | eLife | | | | Plos_genetics | | | |
|---|---|---|---|---|---|---|---|---|---|---|---|---|
| | R-1 | R-2 | R-L | BLEU | R-1 | R-2 | R-L | BLEU | R-1 | R-2 | R-L | BLEU |
| *LLaMA3.1-8B-Instruct* | | | | | | | | | | | | |
| $K = 8$ | 44.74 | 19.18 | 41.91 | 15.5 | 49.28 | 15.02 | 47.02 | 8.56 | 48.76 | **16.79** | 45.06 | **12.09** |
| $K = 16$ | **45.71** | **19.52** | **42.79** | **16.68** | **50.44** | **15.49** | **48.02** | **8.67** | **48.77** | 16.64 | **45.06** | 11.86 |
| $K = 24$ | 45.11 | 19.26 | 42.22 | 15.96 | 49.77 | 15.08 | 47.54 | 8.51 | 48.25 | 16.64 | 45.03 | 11.99 |
| $K = 32$ | 44.02 | 18.78 | 41.32 | 14.28 | 49.37 | 14.94 | 47.13 | 8.27 | 48.49 | 16.55 | 44.67 | 11.78 |

### E.9 Discussion on Generalization of *Magical*

We investigate the generalizability of *Magical* in the following two important scenarios:

**Generalization to unseen datasets within the MLLG task domain**    To explore this critical issue, we introduce a new MLLG dataset named EASI [3]. We selected LLaMA3.1-8B-Instruct as backbone LLM, the experimental settings and result are presented in Table. 12.

Table 12: Generalization performance of *Magical* to unseen datasets within the MLLG task domain. All evaluation metrics are defined such that higher values indicate better performance (↑). The best results are highlighted in **bold**.

| Methods | Tuned | #Params | Train | Test | ROUGE-1 | ROUGE-2 | ROUGE-L | BLEU |
|---|---|---|---|---|---|---|---|---|
| *LLaMA3.1-8B-Instruct* | | | | | | | | |
| Prompt | | | | ESAI | 37.85 | 21.67 | 36.43 | 15.04 |
| LoRA | $A\&B$ | 20M | Cochrane | ESAI | 41.25 | 25.21 | 37.73 | 9.42 |
| LoRA | $A\&B$ | 20M | eLife | ESAI | 38.09 | 23.54 | 19.88 | 6.38 |
| LoRA | $A\&B$ | 20M | Plos_genetics | ESAI | 36.62 | 25.30 | 21.85 | 7.05 |
| LoRA | $A\&B$ | 20M | ESAI | ESAI | 51.74 | 33.01 | 46.82 | 28.09 |
| *Magical* | $B$ | 11M | ESAI | ESAI | **53.82** | **36.55** | **49.87** | **30.57** |

Experimental results show that both prompt-based and LoRA-based methods exhibit poor generalization, performing significantly worse than end-to-end fine-tuning with a newly initialized LoRA on the target dataset. The prompt-based approach suffers from limited performance due to its inability to comprehensively capture the target lay-style. LoRA, on the other hand, is optimized toward a specific lay-style on the training set, and thus demonstrates a marked performance drop when there is a distribution shift between the training and test sets.

We evaluate *Magical* by reusing matrix $A$ trained on Cochrane, eLife, and PLOS Genetics, and introducing a newly initialized matrix $B$, which is fine-tuned on EASI while keeping matrix $A$ frozen. The results show that *Magical* outperforms even the fully fine-tuned LoRA on EASI across all evaluation metrics.

Our key insight is as follows: each heterogeneous MLLG dataset corresponds to a unique lay-style. Current approaches—including prompt-based methods, LoRA, and *Magical*—are not yet capable of robust generalization under truly unseen lay-style conditions.

Nevertheless, we argue that *Magical*'s matrix $A$ constitutes a highly generalizable component. This is because *Magical* explicitly decouples abstractive summarization from lay-style generation. Matrix $A$, which is responsible for summarization, is independent of the lay-style and thus enjoys broad generalizability. Table. 12 is supported by the above empirical results, where *Magical* achieves superior performance and lower parameter overhead compared to LoRA under equivalent fine-tuning settings.

Table 13: Generalization performance of *Magical* to law domain. All evaluation metrics are defined such that higher values indicate better performance (↑). The best results are highlighted in **bold**.

| Methods | #Params | IN-Abs | | | | IN-Ext | | | | UK-Abs | | | |
|---------|---------|--------|------|------|------|--------|------|------|------|--------|------|------|------|
| | | R-1 | R-2 | R-L | BLEU | R-1 | R-2 | R-L | BLEU | R-1 | R-2 | R-L | BLEU |
| *LLaMA3.1-8B-Instruct* | | | | | | | | | | | | | |
| LoRA | 62M | 53.71 | 35.96 | 52.33 | 16.03 | 56.29 | 40.37 | 55.43 | 19.61 | 34.07 | 17.89 | 33.03 | 5.95 |
| rsLoRA | 62M | 57.23 | 38.47 | 55.73 | 19.17 | 56.46 | 41.24 | 55.36 | 21.78 | **38.84** | 21.34 | 37.7 | 7.25 |
| DoRA | 64M | 53.34 | 36.25 | 52.03 | 16.03 | 56.05 | 39.46 | 55.04 | 18.73 | 35.83 | 19.48 | 34.78 | 6.45 |
| PiSSA | 62M | 56.14 | 37.9 | 54.61 | 17.65 | 58.65 | 42.16 | 57.65 | 23.37 | 36.99 | 20.58 | 35.93 | 6.95 |
| *Magical* | 42M | **59.23** | **39.79** | **57.45** | **20.35** | **60.07** | **44.81** | **59.1** | **26.73** | 38.72 | **22.28** | **37.70** | **8.36** |

---

**Expert Text**

Six studies (n = 478) of variable quality were included. A composite outcome of Infant Pain Scale (NIPS), Neonatal Facial Action Coding System (NFCS) and/or Premature Infant Pain Profile (PIPP) score was reported in 288 infants, who did not receive a sweet tasting solution. Meta-analysis showed a significant reduction in the venepuncture versus the heel lance group (SMD -0.76, 95% CI -1.00 to -0.52; I2 = 0%). When a sweet tasting solution was provided the SMD remained significant favouring the venepuncture group (SMD - 0.38, 95% CI -0.69 to -0.07). The typical RD for requiring more than one skin puncture for venepuncture versus heel lance (reported in 4 studies; n = 254) was -0.34 (95% CI -0.43 to -0.25; I2 = 97%). The NNT to avoid one repeat skin puncture was 3 (95% CI 2 to 4). Cry characteristics favoured the venepuncture group but the differences were reduced by the provision of sweet tasting solutions prior to either procedure. Venepuncture, when performed by a skilled phlebotomist, appears to be the method of choice for blood sampling in term neonates. The use of a sweet tasting solution further reduces the pain. Further well designed randomised controlled trials should be conducted in settings where several individuals perform the procedures."

**Ground Truth of Lay Text (Human)**

This review of trials found evidence that venepuncture, when done by a trained practitioner, caused less pain than heel lance. The use of a sweet tasting solution given to the baby prior to the event reduced pain further. The evidence included outcome measures using pain scales, how long the baby cried and how the mother rated their baby's pain.

**LoRA**

This review of six studies (478 babies) found that venepuncture is better than heel lance for blood sampling in term babies. The use of a sweet tasting solution reduces the pain associated with venepuncture. The use of a sweet tasting solution does not reduce the pain associated with heel lance.

*Magical*

This review of six studies (involving 478 newborn babies) found that venepuncture (taking blood from a vein) is less painful than heel lance (pricking the baby's heel) for babies who did not have a sweet tasting solution. When a sweet solution was given to the babies before either procedure, the pain was reduced but venepuncture was still less painful than heel lance. The review also found that venepuncture by a skilled person is less painful than heel lance.

Table 14: Case study on semantic fidelity of LoRA and *Magical* in MLLG task.

**Generalization to Law Domain**   we further introduce a legal summarization task using a widely adopted dataset [61] in the community (IN-Abs, IN-Ext, and UK-Abs). We employ LLaMA3.1-8B-Instruct as the backbone LLM and follow exactly the same experimental settings as used for the MLLG task in the work. The experimental results are presented in Table. 13.

The results show that *Magical* outperforms all baseline methods across most metrics, with the only exception being a slightly lower R-1 score than rsLoRA on the UK-Abs dataset. This demonstrates that *Magical* is also effective in the legal summarization domain.

### E.10 Case Study

**Case in MLLG Tasks**   We randomly select an expert text from the test set to showcase the lay language generation results of LoRA and *Magical*. Notably, LoRA generates the statement: *"The use of a sweet tasting solution does not reduce the pain associated with heel lance,"* which exhibits a notable semantic shift from the original content. In contrast, *Magical* achieves effective lay adaptation while preserving semantic fidelity.

Moreover, for certain domain-specific terms such as "venepuncture" and "heel lance", *Magical* provides further explanatory information that is absent not only in LoRA but even in the ground truth lay text (Human). Such user-friendly explanations significantly enhance the comprehensibility of medical content for end users and facilitate more effective information communication.

**Case in Legal Summarization Tasks**   We further share a similar observation we made in the legal summarization task. In this case, the assertion that "article 134 B does not apply" is the argument of Mr. Tatachari, rather than the conclusion of the High Court. In fact, the High Court concluded that "all the ingredients prescribed by the first column of article 134 B are satisfied." LoRA misinterpreted this distinction, which resulted in a semantic inconsistency. *Magical*, in contrast, successfully accomplished the crucial task of preserving semantic fidelity in MLLG.

---

**Original Legal Case Text**

Appeal No. 801 of 1963.
September 2, 1959 of the MadAppeal by special leave from the judgment and decree dated ras High Court in Second Appeal No. 774 of 1957.
...
On the findings recorded by the High Court, it is clear that the properties belonged to the temple; that they have been transferred by persons who must be deemed to be the previous managers of the temple; and that they have been transferred for valuable consideration. The present suit has been brought against respondents 1 to 3 who are appointed trustees of the temple by respondent No. 4; and so, all the ingredients prescribed by the first column of article 134 B are satisfied. That is why we must reject the ingenious argument urged before us by Mr. Tatachari that article 134 B does not apply to the present case.
...

---

**Summary Generated Text by LoRA**

The appellants filed a suit under section 87 of the Madras Hindu Religious and Charitable Endowments Act, 1951
...
The High Court held that article 134 B does not apply to the present case.
...

---

**Summary Generated Text by *Magical***

The appellants filed a suit against the respondents for a declaration of their title to certain properties
...
The High Court held that all the ingredients prescribed by the first column of article 134 B were satisfied
...

Table 15: Case study on semantic fidelity of LoRA and *Magical* in legal summarization tasks.

