# OpenReview forum: "Magical: Medical Lay Language Generation via Semantic Invariance and Layperson-tailored Adaptation"
_NeurIPS.cc/2025/Conference — NeurIPS 2025 poster_

### Official Review · Reviewer_1PZy · 2025-06-13

**Clarity:** 3
**Significance:** 3
**Originality:** 3
**Rating:** 5
**Confidence:** 4

**Summary:**

This paper addresses the challenge of MLLG, focusing on transforming biomedical content into layman-accessible language. The authors critique the use of standard LoRA in handling semantic fidelity and stylistic diversity across heterogeneous multi-source datasets. They propose Magical, a novel asymmetric LoRA architecture that separates semantic abstraction (via shared matrix A) and lay-style adaptation (via multiple isolated matrices B). The contributions mainly focus on 1) Semantic invariance constraint on matrix A using semantic-relevant layer identification and contrastive learning to preserve semantic fidelity. 2) Layperson-tailored adaptation via dataset-specific B matrices and a recommendation-guided switch for selective matrix activation. 3) Empirical results on three MLLG datasets showing Magical consistently outperforms LoRA and its recent variants in both semantic fidelity and style diversity.

**Questions:**

1. Could Magical be extended to more domains such as multi-lingual lay language generation and other domains including legal simplification or educational content generation, and what changes would be needed to handle such complexity?
2. Have you conducted or considered conducting any human subject evaluations to assess whether generated lay texts are truly easier to understand?
3. How sensitive is Magical’s performance to the choice of the number of B matrices and the top-K semantic layers?

**Ethical Concerns:**

["NO or VERY MINOR ethics concerns only"]

**Final Justification:**

The authors have addressed most of my concerns, particularly regarding generalization to unseen datasets and new domains, human evaluation, hyperparameter selection, and the paper's novelty.

**Limitations:**

The paper does acknowledge that the Recommendation Agent is not implemented, and it also discusses challenges around modeling layperson profiles and proposes future directions such as meta-learning and RL. However, it could be further strengthened by discussing scalability, domain generalizability, data imbalance, and ethical risks caused by hallucination or oversimplification of the generated medical content.

**Paper Formatting Concerns:**

No further formatting concerns.

**Quality:**

4

**Strengths And Weaknesses:**

## Quality
### Strengths:
- Rigorous evaluation across three real-world datasets, including Cochrane, eLife, Plos_genetics.
- Very detailed ablation studies, which effectively isolate component contributions.
### Weaknesses:
- Due to the subjective nature of layman-oriented content comprehension, human evaluation metrics are necessary in this task setting to assess whether the model-generated content is acceptable from the perspective of real-world target users.

## Clarity
### Strengths：
- The framework is well-illustrated through Figure 3 and the proposed methodology and key components are clearly presented in Section 3.2 and 3.3.
### Weaknesses:
- Figure 2 lack detailed captions or axis labels, reducing their standalone interpretability.
- There is a lack of impact analysis of the number of B matrices and the choice of top-K semantic layers.

## Significance
### Strengths：
- The paper addresses a meaningful social need, which is improving public access to biomedical knowledge.
### Weaknesses:
- Despite its significance, the model has limited generalizability beyond MLLG without adaptation, especially since it depends on labeled expert-lay pairs.
- As the provided method is very domain-specific, there is limited discussion on how it can be integrated into existing clinical or public health communication pipelines.
- A main concern is that, while the proposed method is specifically designed within the MLLG domain, many of its core innovations, such as asymmetric low-rank adaptation, semantic invariance constraints, and lay-style routing, are generally applicable to other text generation tasks that require semantic preservation and stylistic transformation. However, the paper currently does not explore this broader applicability. This could make the method appear overly domain-specific, which may limit its perceived impact and usefulness to the broader NLP community.

## Originality
### Strengths：
- It seems to be the first to explicitly decouple semantic abstraction and stylistic transformation in MLLG via asymmetric LoRA.
### Weaknesses:
- While inspired by HydraLoRA, the originality of some technical aspects, such as the use of external switching agents is somewhat incremental.

---

> ### Author Rebuttal · Authors · 2025-07-31
>
> We sincerely appreciate your recognition of the comprehensiveness of our ablation studies, the clarity of our figures and visualizations, and the importance of the addressed task. We have carefully considered the shortcomings you pointed out in our manuscript and provide detailed responses below.
>
> > **Response to Generalization**
>
> We appreciate your suggestion regarding the issue of generalizability. We believe that evaluating the generalization capability of Magical is a crucial step toward improving the quality of our manuscript. Specifically, we investigate the generalizability of Magical in the following two important scenarios:
>
> 1. **Generalization to unseen datasets within the MLLG task domain**
>
> To explore this critical issue, we introduce a **new MLLG dataset** named EASI [1]. We selected `LLaMA3.1-8B-Instruct` as backbone LLM, the experimental settings and result are presented in the table below.
>
> |Methods|tuned modules|#Params|train data|test data|ROUGE-1|ROUGE-2|ROUGE-L|BLEU|
> |-|-|-|-|-|-|-|-|-|
> |Prompt|||||37.85|21.67|36.43|15.04|
> |LoRA|A&B|20M|Cochrane|ESAI|41.25|25.21|37.73|9.42|
> |LoRA|A&B|20M|eLife|ESAI|38.09|23.54|19.88|6.38|
> |LoRA|A&B|20M|Plos_genetics|ESAI|36.62|25.30|21.85|7.05|
> |LoRA|A&B|20M|ESAI|ESAI|51.74|33.01|46.82|28.09|
> |Magical|B|11M|ESAI|ESAI|**53.82**|**36.55**|**49.87**|**30.57**|
>
> Experimental results show that **both prompt-based and LoRA-based methods (Rows 1–4) exhibit poor generalization**, performing significantly worse than end-to-end fine-tuning with a newly initialized LoRA on the target dataset (Row 5). The prompt-based approach suffers from limited performance due to its inability to comprehensively capture the target lay-style. LoRA, on the other hand, is optimized toward a specific lay-style on the training set, and thus demonstrates a marked performance drop when there is a distribution shift between the training and test sets.
>
> We evaluate Magical by reusing matrix A trained on Cochrane, eLife, and PLOS Genetics, and introducing a newly initialized matrix B, which is fine-tuned on EASI while **keeping matrix A frozen** (see Row 6). The results show that **Magical outperforms even the fully fine-tuned LoRA on EASI** (Row 5) across all evaluation metrics.
>
> Our **key insight** is as follows: each heterogeneous MLLG dataset corresponds to a unique lay-style. Current approaches—including prompt-based methods, LoRA, and Magical—are not yet capable of robust generalization under truly unseen lay-style conditions.
>
> Nevertheless, we argue that **Magical’s matrix A constitutes a highly generalizable component**. This is because Magical explicitly decouples abstractive summarization from lay-style generation. Matrix A, which is responsible for summarization, is independent of the lay-style and thus enjoys broad generalizability.
>
> This claim is supported by the above empirical results, where Magical achieves **superior performance** and **lower parameter** overhead compared to LoRA under equivalent fine-tuning settings.
>
> 2. **Generalization to new domains**
>
> Following your suggestion, we further introduce a legal summarization task using a widely adopted dataset [2] in the community (IN-Abs, IN-Ext, and UK-Abs). We employ `LLaMA3.1-8B-Instruct` as the backbone LLM and follow exactly the same experimental settings as used for the MLLG task in the manuscript. The experimental results are presented in the table below.
>
> |Methods|#params|IN-Abs||||IN-Ext||||UK-Abs||||
> |-|-|-|-|-|-|-|-|-|-|-|-|-|-|
> |||R-1|R-2|R-L|BLEU|R-1|R-2|R-L|BLEU|R-1|R-2|R-L|BLEU|
> |LoRA|62M|53.71|35.96|52.33|16.03|56.29|40.37|55.43|19.61|34.07|17.89|33.03|5.95|
> |rsLoRA|62M|57.23|38.47|55.73|19.17|56.46|41.24|55.36|21.78|**38.84**|21.34|37.70|7.25|
> |DoRA|64M|53.34|36.25|52.03|16.03|56.05|39.46|55.04|18.73|35.83|19.48|34.78|6.45|
> |PiSSA|62M|56.14|37.90|54.61|17.65|58.65|42.16|57.65|23.37|36.99|20.58|35.93|6.95|
> |Magical|42M|**59.23**|**39.79**|**57.45**|**20.35**|**60.07**|**44.81**|**59.10**|**26.73**|38.72|**22.28**|**37.70**|**8.36**|
>
> The results show that **Magical outperforms all baseline methods across most metrics**, with the only exception being a slightly lower R-1 score than rsLoRA on the UK-Abs dataset. This demonstrates that Magical is also effective in the legal summarization domain.
>
> > **Response to Human Evaluation**
>
> We fully understand your concern regarding the need for **layperson evaluation** to assess whether the generated texts meet the expectations of target end-users. However, we would like to clarify that evaluating the quality of generated text also crucially involves **verifying factual accuracy** and **content correctness**. For lay users, detecting subtle or domain-specific inaccuracies in the generated lay text can be extremely **challenging**. Therefore, we chose to conduct **expert evaluations** instead of layperson assessments.
>
> We sincerely **apologize** for the space limitations. We **warmly invite** you to refer to our discussion with *Reviewer TVd2* under the section "**Response to External Validations for Automated Metrics**," where we provide **two evaluation methods** for assessing real-world quality, including **human evaluation**.
>
> > **Response to the Analysis of the Number of B Matrices and the Choice of Top-K Semantic Layers**
>
> Thank you for highlighting this point. We agree that analyzing the sensitivity to the choice of top-K layers is important, as it supports our core claim that *different layers of LLMs encode distinct functional representations* (see Line 174 of the manuscript).
>
> Following your suggestion, we have added a sensitivity analysis of the top-K selection based on `LLaMA3.1-8B-Instruct`. The experimental results, shown in the table below, reveal that both overly small values of K (e.g., K = 8) and overly large values (e.g., K = 24 or 32) **result in degraded performance**. This is because selecting too many layers introduces semantic constraints to layers that are not semantically relevant, leading to interference with the LLM’s functional integrity. Conversely, too few layers may fail to impose constraints on all semantically relevant layers, resulting in semantic shift.
>
> |Methods|Cochrane||||eLife||||Plos_genetics||||
> |-|-|-|-|-|-|-|-|-|-|-|-|-|
> ||R-1|R-2|R-L|BLEU|R-1|R-2|R-L|BLEU|R-1|R-2|R-L|BLEU|
> |K=8|44.74|19.18|41.91|15.5|49.28|15.02|47.02|8.56|48.76|**16.79**|45.06|**12.09**|
> |k=16|**45.71**|**19.52**|**42.79**|**16.68**|**50.44**|**15.49**|**48.02**|**8.67**|**48.77**|16.64|**45.06**|11.86|
> |K=24|45.11|19.26|42.22|15.96|49.77|15.08|47.54|8.51|48.25|16.64|45.03|11.99|
> |K=32|44.02|18.78|41.32|14.28|49.37|14.94|47.13|8.27|48.49|16.55|44.67|11.78|
>
> Regarding the number of B matrices, we assign one matrix B per heterogeneous dataset. This is **not treated as a hyperparameter** in our experiments but rather as **a direct reflection of the number of heterogeneous sources involved**.
>
> > **Response to the Clarity of Figure.2**
>
> Thank you for your valuable suggestion. We are highly attentive to the readability and clarity of our manuscript. *In Figure 2, we project the representations of expert text samples and the corresponding generated lay text samples onto the top-2 singular directions of the semantic subspace, which form the x- and y-axes of the KDE plot.* In response, we have revised the caption of Figure 2 to improve clarity and added axis labels to the plot.
>
> > **Response to Novelty**
>
> We sincerely appreciate your valuable feedback and understand your concerns regarding the novelty of the manuscript. We sincerely **apologize** for the space limitations. We **warmly invite** you to refer to our discussion with *Reviewer gwM8* under the section "**Response to Novelty**," where we clarify our perspective on the novelty of Magical.
>
> > **Response to Clinical Application and Limitation Discussion**
>
> Thank you for your valuable suggestion. We believe that expanding the discussion on how Magical can be applied to real-world clinical settings, as well as elaborating on its more limitations, is important. This would greatly enhance the clarity of our work’s practical applicability.
>
> To this end, we offer some **preliminary insights** on how Magical might be translated into practical clinical use. We propose that its deployment follow four key periods:
>
> 1. **Expert-Guided Validation**: Initially, the system should be open to medical experts to ensure the accuracy and reliability of the generated content.
>
> 2. **Lay-Style Generation**: Leveraging Magical’s strengths, it can then produce diverse lay-style text, allowing users to freely select the most suitable ones based on their **latent prefenence**.
>
> 3. **User-Centered Feedback and Learning**: User preferences can be collected in this phase to construct a *Recommendation Agent*.
>
> 4. **Personalized Lay Text Recommendation**: Based on user feedback, the system can ultimately provide precise recommendations of preferred lay-style outputs, facilitating efficient and personalized communication.
>
> In response to your concerns, we have added a **new Discussion section** in the revised manuscript, where we further elaborate on the plans for clinical application, as well as additional limitations of Magical, including those you mentioned such as scalability, generalizability, ethical risks.
>
> Thank you again for your constructive suggestions.
>
> > **References**
>
> [1] Med-EASi: Finely Annotated Dataset and Models for Controllable Simplification of Medical Texts, AAAI, 2023
>
> [2] Legal Case Document Summarization: Extractive and Abstractive Methods and their Evaluation, AACL | IJCNLP，2022
>
> ***
> We sincerely thank you once again for your thoughtful guidance. Your feedback has been immensely valuable to us, and incorporating your suggestions has significantly improved the quality of our manuscript.
> If you have any further questions or concerns, please feel free to contact us at any time. We are always available and look forward to further discussions with you.
>
> Best regards,
>
> All Authors

---

> > ### Comment · Reviewer_1PZy · 2025-08-06
> > **Response to the authors**
> >
> > Thank you for the author's response. The explanations provided have addressed most of my concerns, particularly regarding generalization to unseen datasets and new domains, human evaluation, hyperparameter selection, and the paper's novelty. I hope the authors can incorporate the additional content into the revised manuscript. Accordingly, I have adjusted my score.

---

> ### Author Response · Authors · 2025-08-05
> **Kindly Request for Reviewer's Feedback**
>
> Dear Reviewer 1PZy,
>
> We hope this message finds you well. We completely understand that your schedule may be demanding, and we **sincerely appreciate** the time and effort you have devoted to reviewing our manuscript.
>
> In response to your valuable suggestions, we have made the following clarifications and enhancements to our manuscript:
>
> 1. We investigated **two aspects of the generalization ability of our method**, including its performance on *unseen datasets* and its *transferability to new domains* such as the legal summarization task.
>
> 2. We expanded our evaluation to include both *Large Language Models as Evaluators (Full-scale Evaluation)* and *Human Expert Evaluation (Focused-scale Evaluation)* to assess the **actual perceived quality** of the generated lay texts.
>
> 3. We further included an analysis of our method with respect to the Top-K parameter.
>
> 4. We improved the clarity of Figure 2.
>
> 5. We clarified the novelty of our proposed approach.
>
> 6. We have added further discussion on clinical applications and additional limitations of our work.
>
> We are eager to understand whether our efforts have effectively addressed the limitation in the original manuscript. Given your expertise, **we are eager to engage in further discussion with you**. However, as **the discussion phase draws to a close**, we have yet to receive your response. If possible, we would be deeply grateful if you could spare a moment to share your thoughts, which would be of immense value to us.
>
> Thank you again for your time, thoughtful feedback, and invaluable guidance.
>
> Warmest regards,
>
> All Authors of Submission #22904

---

> ### Author Response · Authors · 2025-08-06
> **Thank you for acknowledging our responses**
>
> Dear Reviewer 1PZy,
>
> Thank you for your careful reading of our manuscript and rebuttal, and for your thoughtful and positive response. Your suggestions are invaluable to us, and we truly appreciate every opportunity to engage in this exchange, which is essential for improving the quality of our work.
>
> Once again, we are sincerely grateful for your kind guidance.
>
> Best wishes,
>
> All Authors of Submission #22904

---

### Official Review · Reviewer_gwM8 · 2025-07-02

**Clarity:** 3
**Significance:** 3
**Originality:** 2
**Rating:** 4
**Confidence:** 3

**Summary:**

This paper introduces Magical, a novel asymmetric LoRA-based architecture designed for Medical Lay Language Generation, a task that involves simplifying complex biomedical texts for non-expert audiences. The authors propose: 1) A shared low-rank matrix A for semantic content modeling (abstractive summarization). 2) Multiple isolated low-rank matrices B for stylistic variation corresponding to heterogeneous lay styles. 3) A Semantic Invariance Constraint applied on matrix A using contrastive learning and semantic-relevant layer identification. 4) A Recommendation-guided Switch that leverages an external agent to dynamically select the appropriate B matrix based on the target lay style. Experiments are conducted across three real-world datasets (Cochrane, eLife, Plos_genetics) and three LLMs (LLaMA3.2-3B, LLaMA3.1-8B, Qwen2.5-7B). Magical achieves state-of-the-art performance compared to various baselines, including rsLoRA, DoRA, PiSSA, and HydraLoRA.

**Questions:**

1 - The proposed asymmetric LoRA framework closely resembles HydraLoRA [24] and Mixture-of-LoRAs [32], both of which use separate low-rank branches and routing mechanisms. Can the authors clarify in more detail how Magical provides a non-trivial architectural or theoretical advancement over these prior approaches? In particular, is there any scenario where HydraLoRA fails but Magical succeeds (beyond empirical results)?


2 - The Recommendation-guided Switch is presented as a key component enabling layperson-tailored adaptation. However, the current implementation relies on manual selection of matrix B. Could the authors elaborate on why a basic learning-based selector (e.g., using dataset metadata, prefix tuning, or retrieval) was not attempted? Additionally, how would performance degrade under imperfect or noisy recommendation in real-world settings?

3 - In Equation (2), the aggregation of BiAx using α_i is central, yet the mechanism for computing α_i during training and inference is under-specified. Are α_i values learned, externally set, or inferred dynamically? If the switch is hard-coded during training, how does this affect generalization to unseen or mixed-style data?

4 - The proposed design is evaluated solely on medical lay language generation. Can the authors provide any evidence or discussion regarding how the method would generalize to other domains requiring stylistic or semantic transformation (e.g., legal summarization, educational simplification)?

**Ethical Concerns:**

["NO or VERY MINOR ethics concerns only"]

**Final Justification:**

After the rebuttal and the discussion with the authors, my final recommendation reflects that the paper's empirical contributions now outweigh its weaknesses, though some concerns remain.

Here is a summary of my final assessment:

* **Strengths (Resolved Issues):** The authors successfully addressed key weaknesses by adding new experiments. The inclusion of BERTScore provides robust quantitative validation for semantic fidelity, and the evaluation on a legal summarization task demonstrates commendable generalization beyond the medical domain. These additions substantially improve the paper's quality.

* **Remaining Concerns:**
    * The core architectural novelty remains incremental, adapting existing Mixture-of-LoRAs frameworks like HydraLoRA to solve a specific, albeit important, problem in MLLG.
    * The "Recommendation-guided Switch," a central component, relies on manual selection. This limits the system's practical autonomy and makes the "layperson-tailored adaptation" a proof-of-concept rather than a fully automated feature.

* **Final Verdict:** The paper is a solid piece of empirical work, and the authors' efforts during the rebuttal have been exemplary. The confirmed strengths and thorough experimentation now justify acceptance. The "Borderline Accept" score reflects the balance between these strong empirical results and the lingering concerns about conceptual novelty and the lack of automation in a key component.

**Limitations:**

Yes

**Quality:**

3

**Strengths And Weaknesses:**

**Strengths**:

1 - MLLG is a high-impact application with societal relevance. The use of PEFT methods such as LoRA (and Magical) enables feasible deployment on limited-resource settings.

2 - The authors perform a solid preliminary investigation showing that multi-source MLLG datasets are highly heterogeneous in structure and style. They identify specific shortcomings of single-LoRA, supporting their architectural decisions with empirical evidence (e.g., Table 1).

3 - Magical achieves significant improvements in both ROUGE and BLEU, while reducing trainable parameters by ~31.66% (Table 2). Moreover, ablation studies (Table 3) are thorough and support the importance of each component.


**Weaknesses**:

1 - The core idea (separate low-rank branches + a selector) is incremental over HydraLoRA [24] and Mixture-of-LoRAs [32]. The only new twist is the semantic contrastive loss on A, whose technical depth is modest.

2 - The Recommendation-guided Switch is central to the claim of “layperson-tailored adaptation,” yet the recommender is manually simulated. As stated in Section 4.1 and 5, “we adopt a manually specified matrix B,” which significantly reduces real-world applicability and novelty

3 - Minor issues: two figures (Figs. 6-7) appear in the appendix but are referenced in Sec. 4 without page cues; typos (“BLUE” for BLEU in multiple tables).

4 - While KDE plots (Fig. 4) are  good qualitative results; no quantitative semantic similarity metric (e.g., BERTScore, CheXplain) is provided to verify “semantic invariance”.

---

> ### Author Rebuttal · Authors · 2025-07-31
>
> We sincerely thank you for acknowledging the importance of the task, the sufficiency of our preliminary investigation, and the comprehensiveness of our experiments. We deeply value the constructive criticisms you raised and respond to each concern in detail below.
>
> > **Response to Novelty (Weakness#1 & Weakness#2)**
>
> We sincerely appreciate your valuable feedback and understand your concerns regarding the novelty of the manuscript. We would like to **clarify** that novelty in academic research should not be limited to proposing disruptive algorithms, but should also encompass the ability to **inspire interest, provoke community discussion, and open up new directions for future work**. In this regard, we argue that Magical achieves meaningful novelty through the following contributions:
>
> 1. Magical provides a comprehensive review of existing approaches and proposes a **new perspective** backed by **empirical evidence**: *“Standard LoRA fails to meet the requirement for semantic fidelity and diverse lay-style generation in the MLLG task.”*
>
> 2. Magical performs **timely adaptations** and responds to the unique challenges faced by LoRA in the MLLG setting by offering a **tailored solution** that directly addresses these shortcomings.
>
> 3. Most importantly, Magical serves as a **catalyst** for rethinking LoRA-based approaches in MLLG, **encouraging the community to explore future directions** for semantic preservation and adaptive lay-style generation.
>
> While we acknowledge that Magical is inspired by HydraLoRA, we highlight that the two methods **differ in their applicability**. In fact, Magical identifies and remedies a limitation of HydraLoRA, which we elaborate on in the next section.
>
> > **Response to Comparison Between HydraLoRA and Magical (Question#1), and on Not Using a Basic Learning-Based Selector(Question#2)**
>
> Thank you for your valuable suggestion.  In response, we revisited the tasks addressed in HydraLoRA, such as *Summarization*, *Closed QA*, and *Information Extraction*.
> It is evident that these tasks are **clearly distinguishable**, and simple prompts are sufficient for an LLM to differentiate among them and select the appropriate matrix B.
> However, when it comes to heterogeneous MLLG tasks, it becomes **challenging to accurately convey the target lay-style using simple prompts** (as discussed in line 227 of our manuscript). This limitation hinders the effective automatic selection of matrix B.
>
> To validate this point, we conducted a detailed experiment in **Appendix E.1: Dilemma of Router-Controlled Selection**, where we adopted HydraLoRA’s original design using a learnable router (i.e., **a basic learning-based selector, as you suggested**). The results demonstrate that the **router fails to learn meaningful selections**—our Figure 5 shows a confusion matrix where the learned routing behavior is indistinguishable from random selection.
>
> Further evidence is provided in Table 3 (“Switch → Router” ablation), where replacing our manual switching with routing leads to a significant drop in performance.
>
> These findings show some **limitations of HydraLoRA**: router-based selection collapses when the task cannot be explicitly described via simple prompts, which is often the case in MLLG. Although Magical uses a simpler, manually defined selection strategy (our **optimal setting** assumes that decisions are made based on a *Recommendation Agent*), **it addresses this core failure effectively**.
>
> > **Response to Sensitivity of Recommendation Agent Performance (Question#2)**
>
> We share your concern about whether the overall performance of Magical is sensitive to the quality of the recommendation agent. In **Appendix E.2: Sensitivity of Recommendation Performance**, we present experiments that simulate reduced accuracy in recommendation. Notably, even when the agent's hit rate drops to 60%, Magical still outperforms standard LoRA.
>
> > **Response to Alpha in Equation (2) and Generalization to Unseen Data (Question#3)**
>
> In Equation (2), the vector $\alpha$ is a binary indicator: the entry corresponding to the selected matrix B is set to 1, while all others are set to 0.
>
> Same as you, we are also deeply concerned with Magical's generalization ability on unseen datasets. To explore this critical issue, we introduce a **new MLLG dataset** named EASI [1]. We selected `LLaMA3.1-8B-Instruct` as backbone LLM, the experimental settings and result are presented in the table below.
>
> |Methods|tuned modules|#Params|train data|test data|ROUGE-1|ROUGE-2|ROUGE-L|BLEU|
> |-|-|-|-|-|-|-|-|-|
> |Prompt|||||37.85|21.67|36.43|15.04|
> |LoRA|A&B|20M|Cochrane|ESAI|41.25|25.21|37.73|9.42|
> |LoRA|A&B|20M|eLife|ESAI|38.09|23.54|19.88|6.38|
> |LoRA|A&B|20M|Plos_genetics|ESAI|36.62|25.30|21.85|7.05|
> |LoRA|A&B|20M|ESAI|ESAI|51.74|33.01|46.82|28.09|
> |Magical|B|11M|ESAI|ESAI|**53.82**|**36.55**|**49.87**|**30.57**|
>
> Experimental results show that **both prompt-based and LoRA-based methods (Rows 1–4) exhibit poor generalization**, performing significantly worse than end-to-end fine-tuning with a newly initialized LoRA on the target dataset (Row 5). The prompt-based approach suffers from limited performance due to its inability to comprehensively capture the target lay-style. LoRA, on the other hand, is optimized toward a specific lay-style on the training set, and thus demonstrates a marked performance drop when there is a distribution shift between the training and test sets.
>
> We evaluate Magical by reusing matrix A trained on Cochrane, eLife, and PLOS Genetics, and introducing a newly initialized matrix B, which is fine-tuned on EASI while **keeping matrix A frozen** (see Row 6). The results show that **Magical outperforms even the fully fine-tuned LoRA on EASI** (Row 5) across all evaluation metrics.
>
> Our **key insight** is as follows: each heterogeneous MLLG dataset corresponds to a unique lay-style. Current approaches—including prompt-based methods, LoRA, and Magical—are not yet capable of robust generalization under truly unseen lay-style conditions.
>
> Nevertheless, we argue that **Magical’s matrix A constitutes a highly generalizable component**. This is because Magical explicitly decouples abstractive summarization from lay-style generation. Matrix A, which is responsible for summarization, is independent of the lay-style and thus enjoys broad generalizability.
>
> This claim is supported by the above empirical results, where Magical achieves **superior performance** and **lower parameter** overhead compared to LoRA under equivalent fine-tuning settings.
>
> > **Response to Writing Issues (Weakness#3)**
>
> We are grateful for your close reading and for pointing out the writing issues in our manuscript. We sincerely apologize for these oversights. In response, we have thoroughly reviewed and revised the manuscript to correct all writing and formatting issues. Thank you again for your careful attention.
>
> > **Response to the Use of a Quantitative Semantic Similarity Metric (Weakness#4)**
>
> We appreciate your valuable suggestion regarding the evaluation of semantic similarity. In response, we have incorporated BERTScore as a complementary evaluation metric. We select `LLaMA3.1-8B-Instruct` as the backbone LLM and conduct our experiments accordingly. The new results are presented in the Table below, and **they align well with the trends we observed using KDE**, thereby further confirming the semantic fidelity of Magical.
>
> |Methods|Cochrane|eLife|Plos_genetics|
> |-|-|-|-|
> ||BertScore|BertScore|BertScore|
> |Prompt|85.81|85.25|86.73|
> |LoRA|88.22|84.50|89.61|
> |rsLoRA|87.73|84.71|88.71|
> |DoRA|88.77|84.71|89.06|
> |PiSSA|89.03|84.83|90.01|
> |Magical|**90.79**|**86.80**|**91.94**|
>
> > **Response to Generalization to Other Domains (Question#4)**
>
> We fully acknowledge your point that evaluating Magical on datasets from other domains is essential for assessing its generalizability.
>
> Following your suggestion, we further introduce a legal summarization task using a widely adopted dataset [2] in the community (IN-Abs, IN-Ext, and UK-Abs). We employ `LLaMA3.1-8B-Instruct` as the backbone LLM and follow exactly the same experimental settings as used for the MLLG task in the manuscript. The experimental results are presented in the table below.
>
> |Methods|#params|IN-Abs||||IN-Ext||||UK-Abs||||
> |-|-|-|-|-|-|-|-|-|-|-|-|-|-|
> |||R-1|R-2|R-L|BLEU|R-1|R-2|R-L|BLEU|R-1|R-2|R-L|BLEU|
> |LoRA|62M|53.71|35.96|52.33|16.03|56.29|40.37|55.43|19.61|34.07|17.89|33.03|5.95|
> |rsLoRA|62M|57.23|38.47|55.73|19.17|56.46|41.24|55.36|21.78|**38.84**|21.34|37.70|7.25|
> |DoRA|64M|53.34|36.25|52.03|16.03|56.05|39.46|55.04|18.73|35.83|19.48|34.78|6.45|
> |PiSSA|62M|56.14|37.90|54.61|17.65|58.65|42.16|57.65|23.37|36.99|20.58|35.93|6.95|
> |Magical|42M|**59.23**|**39.79**|**57.45**|**20.35**|**60.07**|**44.81**|**59.10**|**26.73**|38.72|**22.28**|**37.70**|**8.36**|
>
> The results show that **Magical outperforms all baseline methods across most metrics**, with the only exception being a slightly lower R-1 score than rsLoRA on the UK-Abs dataset. This demonstrates that Magical is also effective in the legal summarization domain.
>
> > **References**
>
> [1] Med-EASi: Finely Annotated Dataset and Models for Controllable Simplification of Medical Texts, AAAI, 2023
>
> [2] Legal Case Document Summarization: Extractive and Abstractive Methods and their Evaluation, AACL | IJCNLP，2022
>
> ***
> We sincerely thank you once again for your thoughtful guidance. Your feedback has been immensely valuable to us, and incorporating your suggestions has significantly improved the quality of our manuscript.
> If you have any further questions or concerns, please feel free to contact us at any time. We are always available and look forward to further discussions with you.
>
> Best regards,
>
> All Authors

---

> > ### Comment · Reviewer_gwM8 · 2025-08-05
> >
> > Thank you for the detailed response to my review. I appreciate the authors' efforts to address each point comprehensively, and I acknowledge the additional experiments, clarification, and expanded evaluation provided.
> >
> > Here are my remarks in response:
> >
> > **On Novelty and Comparison with HydraLoRA / MoLoRA:**
> > I appreciate the clarification that *Magical* is designed to decouple semantic summarization and lay-style generation, and that HydraLoRA’s router struggles in settings like MLLG where task boundaries are fuzzy. However, while the proposed contrastive learning on matrix A and hard switching mechanism are reasonable, I still view these as **modest extensions** rather than fundamentally novel. The paper's main contribution seems architectural and empirical rather than conceptual.
> >
> > **On the Manually-Specified Recommender:**
> > The empirical results do support the claim that router-based selection underperforms in this domain. Nevertheless, the fact remains that Magical relies on **manual selection** of matrix B at inference time, which limits its deployability. The simulation of degraded recommender performance (Appendix E.2) is helpful but does not substitute for a real-world learned selector. As it stands, the Recommendation-guided Switch remains a **conceptual placeholder**.
> >
> > **On Equation (2) and $\alpha_i$ Explanation:**
> > Thank you for clarifying that $\alpha_i$ is a binary indicator. This detail should be **clearly stated** in the main paper, not only inferred from Appendix discussions.
> >
> > **On Domain Generalization:**
> > I appreciate the additional evaluation on legal summarization, which somewhat alleviates concerns about domain specificity.  I would have also appreciated **qualitative analysis** here as well to better assess generalization in stylistic fidelity, not just surface-level metrics.
> >
> > **On Semantic Similarity Metrics:**
> > The inclusion of BERTScore is a welcome improvement!
> >
> >
> > Overall, Magical is a solid and well-executed piece of engineering, addressing a real problem in MLLG. The authors clearly understand the limitations of existing LoRA variants and have carefully adapted known ideas to the unique challenges of heterogeneous, style-sensitive generation. However, the core innovation remains incremental, and the dependency on manually guided switching may weaken claims around practical utility.

---

> ### Author Response · Authors · 2025-08-05
> **Kindly Request for Reviewer's Feedback**
>
> Dear Reviewer gwM8,
>
> We hope this message finds you well. We completely understand that your schedule may be demanding, and we **sincerely appreciate** the time and effort you have devoted to reviewing our manuscript.
>
> In response to your valuable suggestions, we have made the following clarifications and enhancements to our manuscript:
>
> 1. We clarified the novelty of our proposed approach.
>
> 2. We distinguished the applicability of our method from that of HydraLoRA.
>
> 3. We reiterated the sensitivity analysis of our method with respect to the recommendation agents.
>
> 4. We introduced quantitative metrics to evaluate semantic similarity.
>
> 5. We investigated **two aspects of the generalization** ability of our method, including its performance on *unseen datasets* and its *transferability to new domains* such as the legal summarization task.
>
> We are eager to understand whether our efforts have effectively addressed the limitation in the original manuscript. Given your expertise, **we are eager to engage in further discussion with you**. However, as **the discussion phase draws to a close**, we have yet to receive your response. If possible, we would be deeply grateful if you could spare a moment to share your thoughts, which would be of immense value to us.
>
> Thank you again for your time, thoughtful feedback, and invaluable guidance.
>
> Warmest regards,
>
> All Authors of Submission #22904

---

> ### Author Response · Authors · 2025-08-06
> **Kindly Request for Further Discussion with Reviewer gwM8 (1/2)**
>
> Dear Reviewer gwM8,
>
> We sincerely thank the reviewer for their careful reading of our previous rebuttal and for their positive reception of our responses. We are very pleased that the shortcomings in our original manuscript have been fully clarified and amended. We are also very grateful that the reviewer has provided further questions. This indicates to us that our manuscript still has areas for improvement, a point which we take very seriously, and we are eager to engage in further constructive discussion.
>
> > **Response to Novelty**
>
> We completely **understand** and **respect** the reviewer’s decision. We understand that the definition of novelty can be nuanced and acknowledge that this is a matter of seeking common ground while respecting diverse perspectives.
>
> > **Response to Manually-Specified Recommender**
>
> Thank you very much for your suggestion. We would like to offer the following three clarifications:
> 1. We would like to clarify that our **core proposal** is the adoption of a "divide-and-conquer" strategy, which utilizes an external agent to perform the recommendation, thereby avoiding the coupling of lay-style recommendation and lay-style generation within the LoRA. The current manually-specified recommender is intended as a **specific implementation** of this strategy, **rather than the ultimate goal of our methodology**. We highly value this distinction. To prevent any misunderstanding, we describe the conceptual "recommendation agent" in the methodology section (Section 3.3) and then detail this specific implementation in the experimental setup (Section 4.1).
> 2. We wish to clarify that the use of a manual selection process, in lieu of proposed recommendation agent, stems from **a lack of data to model layperson profiles** (as discussed in Section 5 Limitation). We believe this is a limitation not just for our work, but for the MLLG community. **Ideally**, Magical would involve the collection and curation of user data to realize its complete design, including the data-driven recommendation agent. In that state, Magical would undoubtedly be more compelling. However, we must acknowledge the **consensus** that **dataset construction is a long-term endeavor**, which contradicts Magical’s stated purpose as a **timely adjustment**. We think that without a framework like Magical, MLLG research **might** continue to be constrained by the limitations of navie LoRA, potentially leading to a **waste of resources** in the MLLG community.
> 3. We acknowledge what you mentioned, that the current version of the Recommendation-guided Switch is merely a conceptual placeholder. However, we would like to clarify that **making conceptual declarations in methodology while using alternative validation in experiments**, particularly when **limited by contextual factors** such as technology, data, and ethics, is a **common** and **reasonable** scientific practice path in academic development. Magical follows the path: **declaring** the "Recommendation-guided Switch" to replace traditional Router, **implementing** it using a manual recommender as a proxy for the envisioned recommendation agent, and **validating** the correctness of this declaration via ablation studies and sensitivity analysis of the recommender's performance.
>
> > **Response to Practical Utility of Magical**
>
> We thank the reviewer for this important suggestion. We would like to clarify the practical utility of Magical, particularly within the MLLG community where user data on lay-style preferences is currently lacking.
>
> Specifically, we offer some **preliminary insights** on how Magical might be translated into practical clinical use. We propose that its deployment follow four key periods:
>
> 1. **Expert-Guided Validation**: Initially, the system should be open to medical experts to ensure the accuracy and reliability of the generated content.
>
> 2. **Lay-Style Generation**: Leveraging Magical’s strengths, it can then produce diverse lay-style text, allowing users to freely select the most suitable ones based on their **latent prefenence**.
>
> 3. **User-Centered Feedback and Learning**: User preferences can be collected in this phase to construct a *Recommendation Agent*.
>
> 4. **Personalized Lay Text Recommendation**: Based on user feedback, the system can ultimately provide precise recommendations of preferred lay-style outputs, facilitating efficient and personalized communication.
>
> We posit that Magical can be **instrumental in facilitating the collection and construction of this much-needed user lay-style preference data**. In turn, the data-driven recommendation agent, which Magical currently lacks, can be developed and integrated through this process. These two aspects are mutually reinforcing, collectively **promoting the practical application and advancement of the MLLG community**.

---

> ### Author Response · Authors · 2025-08-06
> **Kindly Request for Further Discussion with Reviewer gwM8 (2/2)**
>
> > **Response to the Clarity of alpha**
>
> We thank you for your valuable feedback. We take this comment very seriously and **have revised the manuscript** to clarify this point in the **main manuscript**.
>
> > **Response to Semantic Similarity Metrics**
>
> Thank you very much for your suggestion. We **have revised our manuscript** to incorporate this feedback.
>
> > **Response to Qualitative Analysis**
>
> We share your concern on this matter. In the original manuscript, we presented a case study in Appendix E.4 to illustrate the semantic inconsistencies caused by LoRA, whereas Magical ensures semantic fidelity. We would like to further share a similar observation we made in the legal summarization task.
>
> + Original Legal Case
>
>   >Appeal No. 801 of 1963.
>   >
>   >September 2, 1959 of the MadAppeal by special leave from the judgment and decree dated ras High Court in Second Appeal No. 774 of 1957.
>   >
>   >...
>   >
>   >On the findings recorded by the High Court, it is clear that the properties belonged to the temple; that they have been transferred by persons who must be deemed to be the previous managers of the temple; and that they have been transferred for valuable consideration. The present suit has been brought against respondents 1 to 3 who are appointed trustees of the temple by respondent No. 4; and so, **all the ingredients prescribed by the first column of article 134 B are satisfied**. That is why we must **reject** the ingenious argument urged before us by Mr. Tatachari that **article 134 B does not apply to the present case**.
>   >
>   >...
>
> + Summary Generated Text by LoRA
>
>   >The appellants filed a suit under section 87 of the Madras Hindu Religious and Charitable Endowments Act, 1951
>   >
>   >...
>   >
>   >**The High Court held that article 134 B does not apply to the present case**
>   >
>   >...
>
> + Summary Generated Text by Magical
>
>   >The appellants filed a suit against the respondents for a declaration of their title to certain properties
>   >
>   >...
>   >
>   >**The High Court held that all the ingredients prescribed by the first column of article 134 B were satisfied**
>   >
>   >...
>
> Due to the length of the text (original text: 4572 words; LoRA-generated text: 809 words; Magical-generated text: 664 words), we are presenting only the most salient portions here. The complete cases will be included in the appendix of our revised manuscript.
>
> In this case, the assertion that "article 134 B does not apply" is the argument of Mr. Tatachari, rather than the conclusion of the High Court. In fact, the High Court concluded that “all the ingredients prescribed by the first column of article 134 B are satisfied.” LoRA misinterpreted this distinction, which resulted in a semantic inconsistency. Magical, in contrast, successfully accomplished the crucial task of preserving semantic fidelity in MLLG.
>
> ---
> We would like to express our sincere gratitude to the reviewers for their time and effort in providing a thorough review of our rebuttal. We truly value the insightful comments and suggestions, which have been instrumental in enhancing the quality of our work. Please do not hesitate to contact us if you have any further questions or concerns. We are readily available and look forward to discussing this further with you.
>
> Warmest regards,
>
> All Authors of Submission #22904

---

### Official Review · Reviewer_DTBh · 2025-07-06

**Clarity:** 3
**Significance:** 2
**Originality:** 2
**Rating:** 3
**Confidence:** 2

**Summary:**

This paper introduces Magical, a novel approach to improve how medical texts are simplified for non-experts readers. The authors found that standard LoRA has two critical limitations when applied to medical lay language generation: semantic drift, where LoRA fails to preserve the semantic fidelity during simplification, and style heterogeneity, where different medical datasets require different simplification styles, but standard LoRA struggles to adapt to this diversity. The authors propose use to an asymmetric LoRA architecture to address these issues, using a shared matrix A to handle abstractive summarization while maintaining semantic accuracy, and multiple matrices B to enable adaptation to diverse simplification needs. The authors report Magical outperformed standard LoRA and its variants across three medical datasets (Cochrane, eLife, Plos_genetics).

**Questions:**

The LLMs evaluated in the paper are relatively small (under 8B). Have you benchmarked the performance of stronger models, such as o3-mini or Claude?

**Ethical Concerns:**

["NO or VERY MINOR ethics concerns only"]

**Final Justification:**

Author's rebuttal enhances paper's empirical contributions, I raised my score

**Limitations:**

yes

**Paper Formatting Concerns:**

The font of the title appears to be smaller than the standard.

**Quality:**

3

**Strengths And Weaknesses:**

### Strengths:
* The paper is well-motivated and addressed an important problem - converting complex medical texts into language that non-experts can understand. This is important for improving public health literacy and patient adherence to medical advice.
* The proposed asymmetric LoRA architecture is used in a novel setting.
* The paper is well-written, well-structured, and easy to follow.



### Weakness

* The authors use BLEU and ROUGE as the main metrics to evaluate the quality of medical lay language generation. However, since these metrics rely on surface-form n-gram overlap, they may not capture the true quality of the generated medical lay language. Human evaluation may be needed, and using a strong language model as a judge—such as o3—could also be a reasonable proxy.

* The authors focus on comparing against standard LoRA as the baseline, but a comparison to full-parameter fine-tuning should have been included as well.

* Following my previous two points, I’m not fully convinced by the reported effectiveness of the proposed asymmetric LoRA in medical lay language generation.

---

> ### Author Rebuttal · Authors · 2025-07-31
>
> Thank you for your acknowledgment on the importance of our task and the quality of our manuscript. We are especially grateful for your critical comments, which prompted deep reflection on the limitations of our original submission and motivated us to substantially improve our work. Below, we address your concerns point by point.
>
> > **Response to Evaluation of the Actual Quality of Generated Lay Texts (Weakness#1)**
>
> Thank you for your valuable suggestion. We would like to clarify that **our initial choice of BLEU and ROUGE metrics was based on established practices in prior MLLG literature [1]**, where these metrics have been widely adopted. However, we agree with you that such metrics do not fully reflect the actual perceived quality of generated lay texts.
>
> To address this important limitation, we incorporated two complementary evaluation strategies:
>
> 1. *Large Language Models as Evaluators (Full-scale Evaluation)*:
>
> Following your suggestion, we used `o3-mini` and `DeepSeek-R1` to rate outputs from Magical and all baseline systems on a 1–10 scale. These models acted as automatic evaluators simulating expert judgment across the full test sets. In this experiment, we selected `LLaMA3.1-8B-Instruct` as backbone LLM. As shown in the Table below, **Magical consistently outperformed all baselines across all datasets**, indicating its robust ability to generate high-quality lay texts.
>
> | Methods | Cochrane |  | eLife |  | Plos_genetics |  |
> |---|---|---|---|---|---|---|
> |  | `o3-mini` | `DeepSeek-R1` | `o3-mini` | `DeepSeek-R1` | `o3-mini` | `DeepSeek-R1` |
> | LoRA | 6.57 | 6.44 | 6.94 | 7.02 | 7.35 | 7.43 |
> | rsLoRA | 6.75 | 6.43 | 7.28 | 7.58 | 7.07 | 6.99 |
> | DoRA | 6.77 | 6.56 | 7.19 | 7.66 | 7.11 | 7.28 |
> | PiSSA | 6.54 | 6.33 | 7.30 | 7.36 | 6.83 | 6.87 |
> | Magical | **7.53** | **7.66** | **7.66** | **7.92** | **8.02** | **8.01** |
>
> 2. *Human Expert Evaluation (Focused-scale Evaluation)*:
>
> To further validate Magical’s quality in a real-world setting, we randomly sampled 20 examples from each MLLG dataset (60 samples total), and invited two medical data analysis experts to finish the **human evaluation**. Each expert was shown the expert-level source text, and two lay versions generated by LoRA and Magical—with the generation sources anonymized to ensure fairness. The results, presented in the Table below, show that **both experts consistently rated Magical higher than LoRA**, validating that Magical substantially improves the clarity and quality of generated lay texts.
>
> | Methods | Cochrane (#20) |  | eLife (#20) |  | Plos_genetics (#20) |  |
> |---|---|---|---|---|---|---|
> |  | expert-1 | expert-2 | expert-1 | expert-2 | expert-1 | expert-2 |
> | LoRA | 6.90 | 7.10 | 6.95 | 6.90 | 7.05 | 7.05 |
> | Magical | **8.55** | **8.35** | **7.85** | **7.40** | **8.60** | **8.50** |
>
> We **acknowledge** that the current *Human Expert Evaluation*—based on only 60 samples—has significant **limitations**. We sincerely hope for your understanding that **conducting a comprehensive human evaluation is inherently a long-term effort**. This challenge is further compounded by the length of the data samples used in our study (as shown in Figure 1.a of the original manuscript, each instance averages approximately 300 words), which substantially increases the burden on human experts. Given the constraints of the rebuttal period (one week), it is challenging for us to complete a thorough human evaluation within such a limited timeframe.
>
> We have completed the revision of our manuscript and have incorporated the details of **the human evaluation guidelines**, **the prompts used for LLM-based assessment**, and the **full results** into the updated version.
>
> > **Response to Full-Parameter Fine-Tuning (FFT) Baseline (Weakness#2)**
>
> Thank you for your suggestion. We agree that FFT is an important baseline, as it helps readers better understand why existing MLLG studies tend to choose LoRA over FFT.
>
> In response to your suggestion, we added FFT as a new baseline in our experiments. As shown in the Table below, **Magical consistently outperforms the FFT across all datasets and three different backbone LLMs**. Moreover, FFT introduces **over 150× more trainable parameters** compared to Magical. These results highlight Magical’s dual advantages in performance and parameter efficiency.
>
> | Methods | #params | Cochrane |  |  |  | eLife |  |  |  | Plos_genetics |  |  |  |
> |---|---|---|---|---|---|---|---|---|---|---|---|---|---|
> |  |  | R-1 | R-2 | R-L | BLEU | R-1 | R-2 | R-L | BLEU | R-1 | R-2 | R-L | BLEU |
> |`LLaMA3.2-3B-Instruct`|  |  |  |  |  |  |  |  |  |  |  |  |  |
> | FFT | 3,606M | 44.19 | 18.28 | 41.37 | 13.84 | 47.01 | 12.72 | 45.01 | 7.20 | 43.5 | 14.12 | 40.25 | 7.79 |
> | Magical | 24M | **45.33** | **19.39** | **42.36** | **16.66** | **49.16** | **14.68** | **46.91** | **8.30** | **47.50** | **15.47** | **44.03** | **10.24** |
> | `LLaMA3.1-8B-Instruct` |  |  |  |  |  |  |  |  |  |  |  |  |  |
> | FFT | 8,030M | 43.2 | 18.34 | 40.27 | 15.90 | 48.76 | 13.30 | 46.92 | 7.82 | 44.81 | 13.99 | 41.61 | 7.99 |
> | Magical | 42M | **45.71** | **19.52** | **42.79** | **16.68** | **50.44** | **15.49** | **48.02** | **8.67** | **48.77** | **16.64** | **45.06** | **11.86** |
> | `Qwen2.5-7B-Instruct` |  |  |  |  |  |  |  |  |  |  |  |  |  |
> | FFT | 7,615M | 45.99 | 19.62 | 42.58 | 15.81 | 47.73 | 14.49 | 45.36 | 7.20 | 45.49 | 15.64 | 43.20 | 9.97 |
> | Magical | 42M | **47.42** | **20.81** | **44.38** | **17.89** | **50.50** | **15.28** | **48.16** | **8.66** | **48.54** | **16.39** | **44.79** | **11.42** |
>
> Our **perspective on FFT *vs.* LoRA** is as follows:
> We actually considered using FFT for SFT, yet LoRA are a more **economic and effective** choice for MLLG task. The reason is that for MLLG, we often do not have a large amount of training data, and the FFT strategy may lead to **overfitting** (e.g., learning to respond with some specific tokens instead of learning what the style really depicts). Research works like [2,3,4] also demonstrate that FFT suffers from overfitting and cannot perform as good as other PEFT methods. Moreover, the relatively high training cost of FFT may pose a bottleneck in terms of **training efficiency**.
> Therefore, taking both training efficiency and effectiveness into consideration, we choose to apply LoRA as the optimization algorithm for SFT.
>
> > **Response to Large-Scale LLMs as Baselines (Question#1)**
>
> Thank you for your suggestion. We believe that adding a large-scale LLM such as `o3-mini` as a new baseline would help readers better appreciate the performance advantages of Magical.
>
> Following your suggestion, we added `o3-mini` as a new baseline and applied the prompt-based method described in Section 4.1 of the original manuscript. As shown in the Table below, **the performance of `o3-mini` with prompting was suboptimal**, consistent with the conclusions of our original submission. We argue that simple prompt strategies are insufficient to comprehensively capture the target lay-style, making non-learning-based approaches poorly adaptable to the target dataset. This observation is also reported in recent literature. For example, [5] presents results showing that LoRA-based fine-tuned LLaMA-2 outperforms GPT-4.
>
> | Methods | #params | Cochrane |  |  |  | eLife |  |  |  | Plos_genetics |  |  |  |
> |---|---|---|---|---|---|---|---|---|---|---|---|---|---|
> |  |  | R-1 | R-2 | R-L | BLEU | R-1 | R-2 | R-L | BLEU | R-1 | R-2 | R-L | BLEU |
> | `o3-mini` | N/A | 46.50 | 14.30 | 42.82 | 9.00 | 38.15 | 9.48 | 34.75 | 2.46 | 42.59 | 10.42 | 38.76 | 5.65 |
> | Magical | 42M | **47.42** | **20.81** | **44.38** | **17.89** | **50.50** | **15.28** | **48.16** | **8.66** | **48.54** | **16.39** | **44.79** | **11.42** |
>
> > **Response to Formatting Issues**
>
> We greatly appreciate your attention to detail. We sincerely apologize for the formatting errors caused by our oversight. In response, we carefully re-checked our full manuscript against the official NeurIPS formatting guidelines and have corrected all inconsistencies. We take this issue seriously and thank you again for bringing it to our attention.
>
> > **References**
>
> [1] A dataset for plain language adaptation of biomedical abstracts. Nature Scientific Data, 2023
>
> [2] Low-rank adaptation for multilingual summarization: An empirical study, NAACL, 2023
>
> [3] Parameter-efficient orthogonal finetuning via butterfly factorization. ICLR, 2024
>
> [4] AdaLoRA: Adaptive budget allocation for parameter-efficient fine-tuning. ICLR, 2023
>
> [5] Retrieval augmentation of large language models for lay language generation. Journal of Biomedical Informatics, 2024
>
> ***
> We sincerely thank you once again for your thoughtful guidance. Your feedback has been immensely valuable to us, and incorporating your suggestions has significantly improved the quality of our manuscript.
> If you have any further questions or concerns, please feel free to contact us at any time. We are always available and look forward to further discussions with you.
>
> Best regards,
>
> All Authors

---

> ### Author Response · Authors · 2025-08-05
> **Kindly Request for Reviewer's Feedback**
>
> Dear Reviewer DTBh,
>
> We hope this message finds you well. We completely understand that your schedule may be demanding, and we **sincerely appreciate** the time and effort you have devoted to reviewing our manuscript.
>
> In response to your thoughtful suggestions and questions, we have carefully prepared **a detailed rebuttal that we hope addresses your concerns**.
>
> Specifically, we have made the following improvements:
>
> 1. We expanded our evaluation to include both *Large Language Models as Evaluators (Full-scale Evaluation)* and *Human Expert Evaluation (Focused-scale Evaluation)* to assess the **actual perceived quality** of the generated lay texts. Results consistently **demonstrate the superiority of our method** under both evaluation protocols.
>
> 2. Following your suggestion, we introduced new baselines including *Full-Parameter Fine-Tuning* and *Large-Scale LLMs* (e.g., o3-mini). Experimental results show that **our approach outperforms these strong baselines**, and we further provide **insightful analysis into their limitations**.
>
> We are eager to understand whether our efforts have effectively addressed the limitation in the original manuscript. Given your expertise, **we are eager to engage in further discussion with you**. However, as **the discussion phase draws to a close**, we have yet to receive your response. If possible, we would be deeply grateful if you could spare a moment to share your thoughts, which would be of immense value to us.
>
> Thank you again for your time, thoughtful feedback, and invaluable guidance.
>
> Warmest regards,
>
> All Authors of Submission #22904

---

> > ### Comment · Area_Chair_6vPY · 2025-08-06
> >
> > Hi reviewer,
> >
> > Thanks for your previous hard work in the review phase. Now, we need to perform the next step to discuss this article for the decision on whether it can be accepted. Please feel free to read the other reviews, and actively participate in discussions if you have a different opinion. Thanks for your contributions for our NeurIPS community again.
> >
> > Best,
> > AC

---

> ### Author Response · Authors · 2025-08-08
> **A kind reminder as the deadline is approaching**
>
> Dear Reviewer DTBh,
>
> We hope this message finds you well.
>
> We are writing to kindly remind you that the deadline for the discussion phase is drawing close, with **only one day remaining**. We **greatly appreciate** the time and effort you have already dedicated to reviewing our work, and we have provided point-by-point responses to your constructive suggestions and questions.
>
> However, it appears that we **have not yet** received your response to our rebuttal, and we are wondering if our responses have addressed your concerns.
>
> Your insights and feedback are invaluable to us, and we **sincerely hope** you can **participate in the discussion phase** so that we can **together** promote a review-author discussion environment characterized by fairness, politeness, and calmness, as encouraged by the NeurIPS Program Chairs, and foster a more positive NeurIPS community.
>
> Thank you once again for your time and consideration. We are looking forward to hearing from you soon.
>
> Warmest regards,
>
> All Authors of Submission #22904

---

### Official Review · Reviewer_TVd2 · 2025-07-07

**Clarity:** 3
**Significance:** 4
**Originality:** 3
**Rating:** 5
**Confidence:** 4

**Summary:**

The authors aim to address the limitations of style transfer in LoRA models, particularly in the medical domain for expert to lay styles. The goal of th paper is to specifically address heterogenous datasets where the content can be quite different. The proposed method, "Magical", implements LoRA with a single A matrix and multiple B matrices for each different style or dataset.

The authors evaluate this method on 3 biomedical corpora across three backbone LLMs and shows performance improvements on top of vanilla LoRA.

**Questions:**

Are there external validations for the automated metrics? I understand the need for deterministic ways to evaluate quality, but how do we know the proposed methods do not just overfit to the measured quantity vs. actual perceived quality? For example, a small panel of physicians and patients evaluating quality.

**Ethical Concerns:**

["NO or VERY MINOR ethics concerns only"]

**Limitations:**

Yes

**Quality:**

3

**Strengths And Weaknesses:**

This is a useful and interesting paper that other researchers will appreciate reading, and represents a positive step in an area with limited attention, namely, tackling heterogeneous biomedical datasets. The method itself can be implemented with relatively few steps, and the experiments are thorough.

There are some limitations with how realistic the evaluations are, given the static nature and limited scope of the datasets, but these are general problems with medical LLM evaluation.

---

> ### Author Rebuttal · Authors · 2025-07-31
>
> Thank you for your acknowledgment on the usefulness of our method, the importance of the task, the thoroughness of our experiments, and the potential to engage a wider audience. We are particularly grateful for your clarification that the limitations of evaluation due to static and narrow datasets are a well-known and general challenge in the field of medical LLMs. Your perspective is highly encouraging and has motivated us to further advance the task of MLLG.
>
> > **Response to Evaluation Realism Concerns (Weaknesses#1)**
>
> We appreciate your insightful observation. We **also acknowledge** the inherent limitations posed by the static nature and limited scope of the datasets, which affect the realism and generalizability of LLM evaluation. As you noted, this remains a **general problem** across the medical NLP community.
> Despite these challenges, we adhered to **standard academic protocols** by employing benchmark datasets [1,2,3] and evaluation metrics [4] that are **widely used in prior research**. In the manuscript, we aims to **present our observations and insights** to inspire further discussion within the community. We would like to **emphasize** that, within such a constrained setting, **Magical makes a modest yet meaningful step forward**.
>
> > **Response to External Validations for Automated Metrics (Question#1)**
>
> We appreciate your raising the critical question: *“How do we know the proposed methods do not just overfit to the measured quantity vs. actual perceived quality?”*—a fundamental concern in evaluating MLLG systems. To address this,
> We conducted a **human evaluation** study and invited two experts specializing in medical data analysis. We randomly selected 20 samples from each MLLG dataset (totaling 60 samples) and asked the experts to rate the quality of generated lay summaries on a scale from 1 to 10. Each expert was presented with the original expert-level text and two lay versions generated by LoRA and Magical, respectively, in a blind setting (i.e., the sources of the lay summaries were not disclosed).
>
> We select `LLaMA3.1-8B-Instruct` as backbone LLM and the evaluation results are summarized in the table below. **Both experts consistently rated Magical significantly higher, confirming its superiority in producing high-quality, understandable lay summaries**.
>
> | Methods | Cochrane (#20) |  | eLife (#20) |  | Plos_genetics (#20) |  |
> |---|---|---|---|---|---|---|
> |  | expert-1 | expert-2 | expert-1 | expert-2 | expert-1 | expert-2 |
> | LoRA | 6.90 | 7.10 | 6.95 | 6.90 | 7.05 | 7.05 |
> | Magical | **8.55** | **8.35** | **7.85** | **7.40** | **8.60** | **8.50** |
>
> We **acknowledge** that the current evaluation—based on only 60 samples—has significant **limitations**. We sincerely hope for your understanding that **conducting a comprehensive human evaluation is inherently a long-term effort**. This challenge is further compounded by the length of the data samples used in our study (as shown in Figure 1.a of the original manuscript, each instance averages approximately 300 words), which substantially increases the burden on human experts. Given the constraints of the rebuttal period (one week), it is challenging for us to complete a thorough human evaluation within such a limited timeframe.
>
> To complement this, we additionally employed `o3-mini` and `DeepSeek-R1` as **LLM-based evaluators** to simulate large-scale expert-like judgments. These results, presented in the table below, consistently show that **Magical outperforms all baselines across all datasets**, further supporting that its improvements are not merely due to overfitting to specific metrics, but reflect genuine quality gains.
>
> | Methods | Cochrane |  | eLife |  | Plos_genetics |  |
> |---|---|---|---|---|---|---|
> |  | `o3-mini` | `DeepSeek-R1` | `o3-mini` | `DeepSeek-R1` | `o3-mini` | `DeepSeek-R1` |
> | LoRA | 6.57 | 6.44 | 6.94 | 7.02 | 7.35 | 7.43 |
> | rsLoRA | 6.75 | 6.43 | 7.28 | 7.58 | 7.07 | 6.99 |
> | DoRA | 6.77 | 6.56 | 7.19 | 7.66 | 7.11 | 7.28 |
> | PiSSA | 6.54 | 6.33 | 7.30 | 7.36 | 6.83 | 6.87 |
> | Magical | **7.53** | **7.66** | **7.66** | **7.92** | **8.02** | **8.01** |
>
> We have completed the revision of our manuscript and have incorporated the details of **the human evaluation guidelines**, **the prompts used for LLM-based assessment**, and the **full results** into the updated version.
>
> > **References**
>
> [1] Paragraph-level Simplification of Medical Texts, NAACL, 2021
>
> [2] Making Science Simple: Corpora for the Lay Summarisation of Scientific Literature, EMNLP, 2022
>
> [3] Retrieval augmentation of large language models for lay language generation, Journal of Biomedical Informatics, 2023
>
> [4] A dataset for plain language adaptation of biomedical abstracts. Nature Scientific Data, 2023
>
> ***
> Once again, we deeply appreciate your constructive feedback. Your emphasis on perceived quality evaluation is crucial for MLLG and has inspired us to push further toward building methods with real-world impact and user value.
> If you have any further questions or concerns, please feel free to contact us at any time. We are always available and look forward to further discussions with you.
>
> Best regards,
>
> All Authors

---

> > ### Comment · Reviewer_TVd2 · 2025-08-06
> >
> > Thanks for your response and for the updated information. I will confirm my positive score and recommend publication.

---

> ### Author Response · Authors · 2025-08-05
> **Kindly Request for Reviewer's Feedback**
>
> Dear Reviewer TVd2,
>
> We hope this message finds you well. We completely understand that your schedule may be demanding, and we **sincerely appreciate** the time and effort you have devoted to reviewing our manuscript.
>
> In response to your thoughtful suggestions and questions, we have carefully prepared **a detailed rebuttal that we hope addresses your concerns**.
>
> Specifically, following your valuable feedback, we conducted **two evaluations** to assess the perceived quality of our method:
>
> 1. *Human Expert Evaluation (Focused-scale Evaluation)*: As suggested, we invited two domain experts to manually assess a randomly selected subset of our dataset, enabling a precise and qualitative examination.
>
> 2. *Large Language Models as Evaluators (Full-scale Evaluation)*: To compensate for the limited scale of human evaluation, we employed two large language models as judges to conduct a comprehensive assessment over the entire dataset.
>
> Experimental results indicate that **our method enhances actual perceived quality** rather than merely optimizing for superficial metric gains.
>
> We are eager to understand whether our efforts have effectively addressed the lack of evaluation on actual perceived quality in the original manuscript. Given your expertise, **we are eager to engage in further discussion with you**. However, as **the discussion phase draws to a close**, we have yet to receive your response. If possible, we would be deeply grateful if you could spare a moment to share your thoughts, which would be of immense value to us.
>
> Thank you again for your time, thoughtful feedback, and invaluable guidance.
>
> Warmest regards,
>
> All Authors of Submission #22904

---

> > ### Comment · Area_Chair_6vPY · 2025-08-06
> >
> > Hi reviewer,
> >
> > Thanks for your previous hard work in the review phase. Now, we need to perform the next step to discuss this article for the decision on whether it can be accepted. Please feel free to read the other reviews, and actively participate in discussions if you have a different opinion. Thanks for your contributions for our NeurIPS community again.
> >
> > Best,
> > AC

---

> ### Author Response · Authors · 2025-08-07
> **Thank you for acknowledging our responses**
>
> Dear Reviewer TVd2,
>
> Thank you for your careful reading of our manuscript and rebuttal, and for your thoughtful and positive response. Your suggestions are invaluable to us, and we truly appreciate every opportunity to engage in this exchange, which is essential for improving the quality of our work.
>
> Once again, we are sincerely grateful for your kind guidance.
>
> Best wishes,
>
> All Authors of Submission #22904

---

### Author Response · Authors · 2025-08-09
**General Rebuttal**

Dear Program Chairs, Senior Area Chairs, Area Chairs, and Reviewers,

We would like to express our sincere gratitude to all reviewers for their positive recognition of our work and for their insightful and constructive feedback.

In particular, we appreciate that **all reviewers** acknowledged the **importance of the MLLG task**. We thank Reviewer TVd2 for acknowledging our method as a **positive step forward** on MLLG, Reviewer DTBh for acknowledging the **well-motivated method** and **quality of our manuscript writing**, Reviewer gwM8 for acknowledging our method’s suitability for **deployment under limited resources**, **solid preliminary investigation**, and **significant improvements**, and Reviewer 1PZy for acknowledging our **detailed experiments**, the **clarity of our manuscript**.

We also value the insightful comments and constructive suggestions, which have been instrumental in improving the quality of our work. During the rebuttal period, we have provided the following responses and additions:

1. **Human Evaluation and LLMs as Judges**: Following the suggestions of Reviewers TVd2, DTBh, and 1PZy, we supplemented our evaluation with both human assessments and LLM-based judgments to measure the actual perceived quality of Magical’s outputs. This addition was **recognized positively by Reviewers TVd2 and 1PZy**.

2. **Additional Baselines**: In line with Reviewer DTBh’s suggestion, we included Full-Parameter Fine-Tuning and Large-Scale LLMs as new baselines. Experimental results, along with prior literature, **confirm the superiority of our method**.

3. **Novelty Clarification**: In response to the novelty-related concerns raised by Reviewers gwM8 and 1PZy, we clarified the novelty of Magical, and this clarification was **positively acknowledged by Reviewer 1PZy**.

4. **Design Rationale of Magical**: Responding to Reviewer gwM8’s request, we revisited the tasks involved in HydraLoRA and re-presented experiments from our original supplementary material to elucidate Magical’s design rationale. **Reviewer gwM8 noted that Magical clearly understands the limitations of existing LoRA variants and adapts to address the unique challenges of MLLG.**

5. **Generalization Ability**: Following the suggestions of Reviewers gwM8 and 1PZy, we evaluated Magical’s generalization in two scenarios. **Both reviewers explicitly stated that these additions resolved their concerns.**

6. **Semantic-Invariance Quantitative Evaluation**: As suggested by Reviewer gwM8, we used BERTScore to assess the semantic invariance of Magical, which was positively **acknowledged by Reviewer gwM8**.

7. **Hyperparameter Analysis**: Following Reviewer 1PZy’s suggestion, we added a sensitivity analysis for the Top-K parameter and clarified the experimental settings for the number of matrices B. This response was **recognized by Reviewer 1PZy.**

8. **Clinical Applications and Limitation Discussion**: Addressing Reviewer 1PZy’s suggestion, we provided preliminary considerations for the clinical application of Magical and discussed its limitations regarding scalability, generalizability, and ethical risks.

For the additional experiments above, we have provided further analysis and insights in our detailed responses, and we confirm that **the new results continue to support the primary conclusions of our original manuscript.** We commit to incorporating all **the above additions into the revised manuscript** to further improve its quality.

Once again, we sincerely thank the Program Chairs, Senior Area Chairs, Area Chairs, and Reviewers for your time and dedication in reviewing our submission.

Best regards,

All authors of submission 22904

---

### Decision · Program_Chairs · 2025-09-17

**Decision:**

Accept (poster)

**Comment:**

The paper addresses the important problem of Medical Lay Language Generation (MLLG), which seeks to make complex biomedical texts understandable for non-expert audiences, with direct implications for health literacy and clinical communication. The authors identify limitations of existing parameter-efficient fine-tuning methods, particularly standard LoRA, which struggle to simultaneously preserve semantic fidelity and adapt to heterogeneous lay styles across datasets. To address this, they propose Magical, an asymmetric LoRA framework that employs a shared matrix for abstractive summarization, isolated matrices for diverse lay-style generation, and a semantic invariance constraint to ensure content fidelity. In addition, a recommendation-guided switching mechanism is introduced to better adapt to style variation. Experiments on three medical datasets across multiple LLM backbones demonstrate consistent performance improvements over standard LoRA and several of its variants, with the added benefit of parameter efficiency.

The strengths of the paper lie in its timely and socially impactful application domain, the solid empirical investigation demonstrating the inadequacies of existing approaches, and the comprehensive experimental validation across datasets, backbones, and baselines. The method is straightforward to implement yet effective, showing both performance gains and reduced parameter costs. Reviewers noted that the paper is clearly written and accessible, making it an engaging contribution for the broader community. The rebuttal phase further strengthened the work: the authors responded constructively by incorporating human evaluation and LLM-based assessments to validate perceived quality, adding Full-Parameter Fine-Tuning and large-scale LLMs (e.g., o3-mini) as baselines, and introducing semantic similarity metrics (BERTScore). They also expanded the scope of evaluation to a legal summarization dataset, demonstrating generalization beyond the medical domain. These additions address the major concerns raised in the initial reviews.

The weaknesses primarily concern the incremental nature of the architectural contribution relative to HydraLoRA and Mixture-of-LoRAs, and the fact that the recommendation-guided switch currently relies on manual selection rather than a fully automated recommender. While these are valid points, the authors convincingly argue that router-based approaches fail in the fuzzy task boundaries of MLLG, and that Magical’s decoupling of summarization and style generation provides a pragmatic and empirically validated solution. The manual switch is acknowledged as a conceptual placeholder, and sensitivity analysis shows that even under degraded recommendation performance, Magical continues to outperform baselines. Thus, while the novelty is more empirical than conceptual, the contribution is nonetheless valuable and timely.

During the rebuttal and discussion, reviewers’ concerns about evaluation realism, semantic fidelity, generalization, and baseline completeness were addressed through new experiments and expanded analyses. Reviewers who initially had reservations either maintained or raised their recommendations after these clarifications, with some explicitly confirming that their concerns were resolved. The overall consensus is that the empirical contributions are strong, the problem setting is important, and the paper’s improvements are meaningful for both research and real-world applications.

In summary, despite some incremental aspects, this paper makes a well-executed and timely contribution to a high-impact problem. Its thorough experimentation, careful analysis, and strong rebuttal efforts justify acceptance. While it may not merit a spotlight due to modest architectural novelty, it stands as a solid and valuable addition to the conference program, and I recommend acceptance.